# SPARSE FEATURE CIRCUITS: DISCOVERING AND EDITING INTERPRETABLE CAUSAL GRAPHS IN LANGUAGE MODELS

**Samuel Marks**[*]
Northeastern University

**Can Rager**
Independent

**Eric J. Michaud**
MIT

**Yonatan Belinkov**
Technion – IIT

**David Bau**
Northeastern University

**Aaron Mueller**[*]
Northeastern University

## ABSTRACT

We introduce methods for discovering and applying **sparse feature circuits**. These are causally implicated subnetworks of human-interpretable features for explaining language model behaviors. Circuits identified in prior work consist of polysemantic and difficult-to-interpret units like attention heads or neurons, rendering them unsuitable for many downstream applications. In contrast, sparse feature circuits enable detailed understanding of unanticipated mechanisms in neural networks. Because they are based on fine-grained units, sparse feature circuits are useful for downstream tasks: We introduce SHIFT, where we improve the generalization of a classifier by ablating features that a human judges to be task-irrelevant. Finally, we demonstrate an entirely unsupervised and scalable interpretability pipeline by discovering thousands of sparse feature circuits for automatically discovered model behaviors.

## 1 INTRODUCTION

The key challenge of interpretability research is to scalably explain the many unanticipated behaviors of neural networks (NNs). Much recent work explains NN behaviors in terms of coarse-grained model components, for example by implicating certain induction heads in in-context learning (Olsson et al., 2022) or MLP modules in factual recall (Meng et al., 2022; Geva et al., 2023; Nanda et al., 2023, *inter alia*). However, such components are generally polysemantic (Elhage et al., 2022) and hard to interpret, making it difficult to apply mechanistic insights to downstream applications. On the other hand, prior methods for analyzing behaviors in terms of fine-grained units (Kim et al., 2018; Belinkov, 2022; Geiger et al., 2023; Zou et al., 2023) attempt to fit model internals to researcher-specified mechanistic hypotheses using researcher-curated data. These approaches are not well-suited to the many cases where researchers cannot anticipate ahead of time how models internally implement their surprising behaviors.

We propose to explain model behaviors using fine-grained components that play narrow, interpretable roles. Doing so requires us to address two challenges: First, we must identify an appropriate fine-grained unit of analysis, since obvious choices like neurons[1] are rarely interpretable, and units discovered via supervised methods like linear probing require pre-existing hypotheses (Mueller et al., 2024). Second, we must address the scalability problem posed by searching for causal circuits over a large number of fine-grained units.

We leverage recent progress in dictionary learning for NN interpretability (Bricken et al., 2023; Cunningham et al., 2024) to tackle the first challenge. Namely, we use **sparse autoencoders** (SAEs) to identify directions in LM latent spaces which represent human-interpretable concepts. Then, to address the scalability challenge, we employ linear approximations (Sundararajan et al., 2017; Nanda,

---

[*]Correspondence to `s.marks@northeastern.edu` and `aa.mueller@northeastern.edu`.

[1]We use "neuron" to refer to a basis-aligned direction in an LM's latent space (not necessarily preceded by a nonlinearity).

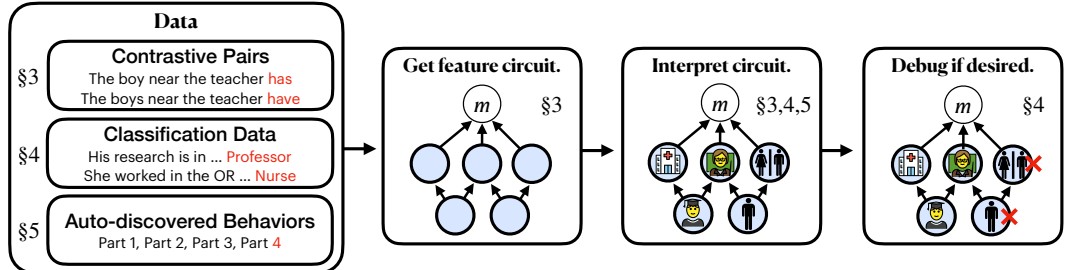

Figure 1: Overview. Given contrastive input pairs, classification data, or automatically discovered model behaviors, we discover circuits composed of human-interpretable sparse features to explain their underlying mechanisms. We then label each feature according to what it activates on or causes the model to predict. Finally, if desired, we can ablate spurious features out of the circuit to modify how the system generalizes.

2022; Syed et al., 2023) to efficiently identify SAE features which are most causally implicated in model behaviors, as well as connections between these features. The result is a **sparse feature circuit** which explains how model behaviors arise via interactions among fine-grained human-interpretable units.

Sparse feature circuits can be productively used in downstream applications. We introduce a technique, Sparse Human-Interpretable Feature Trimming (SHIFT; §4), which shifts the generalization of an LM classifier by surgically removing sensitivity to unintended signals. Unlike previous work on spurious cue removal—which isolates spurious signals using disambiguating data—SHIFT identifies unintended signals using interpretability and human judgement. We thus showcase SHIFT by debiasing a classifier in a worst-case setting, where an unintended signal (gender) is *perfectly predictive* of target labels (profession).

Finally, we demonstrate our method's scalability by automatically discovering thousands of narrow LM behaviors—for example, predicting "to" as an infinitive object or predicting commas in dates—with the clustering approach of Michaud et al. (2023), and then automatically discovering feature circuits for these behaviors (§5).

Our contributions are summarized as follows (Figure 1):

1. A scalable method to discover sparse **feature circuits**. We validate our method by discovering and evaluating feature circuits on a suite of subject-verb agreement tasks.
2. **SHIFT**, a technique for removing a LM classifier's sensitivity to unintended signals, even without data that isolate these signals.
3. A *fully-unsupervised* pipeline for computing feature circuits for thousands of automatically discovered LM behaviors, viewable at `feature-circuits.xyz`.

We release code, data and autoencoders at `github.com/saprmarks/feature-circuits`.

## 2 FORMULATION

**Feature disentanglement with sparse autoencoders.** A fundamental challenge in NN interpretability is that individual neurons are rarely interpretable (Elhage et al., 2022). Therefore, many interpretability researchers have recently turned to **sparse autoencoders** (SAEs), an unsupervised technique for identifying a large number of interpretable NN latents (Cunningham et al., 2024; Bricken et al., 2023; Templeton et al., 2024; Rajamanoharan et al., 2024a;b). Given a model component with latent space $\mathbb{R}^d$ and an activation $\mathbf{x} \in \mathbb{R}^d$, an SAE computes a decomposition

$$\mathbf{x} = \hat{\mathbf{x}} + \boldsymbol{\epsilon}(\mathbf{x}) = \sum_{i=1}^{d_{\text{SAE}}} f_i(\mathbf{x})\mathbf{v}_i + \mathbf{b} + \boldsymbol{\epsilon}(\mathbf{x}) \tag{1}$$

into an approximate reconstruction $\hat{\mathbf{x}}$ as a sparse sum of features $\mathbf{v}_i$ and an *SAE error term* $\boldsymbol{\epsilon}(\mathbf{x}) \in \mathbb{R}^d$. Here $d_{\text{SAE}}$ is the *width* of the SAE, the *features* $\mathbf{v}_i \in \mathbb{R}^d$ are unit vectors, the *feature activations* $f_i(\mathbf{x}) \geq 0$ are a sparse set of coefficients, and $\mathbf{b} \in \mathbb{R}^d$ is a bias. SAEs are trained on an objective

which promotes having a small reconstruction error $\|\mathbf{x} - \hat{\mathbf{x}}\|_2$ while using only a sparse set of feature activations $f_i(\mathbf{x})$. Rather than discard the error terms $\epsilon$ for the purposes of circuit discovery, our methods handle them gracefully by incorporating them into our sparse feature circuits; this gives a principled decomposition of model behaviors into contributions from interpretable features and error components not yet captured by our SAEs.

In this work, we leverage the following suites of SAEs:

- A suite of SAEs we train for each sublayer (attention layer, MLP, residual stream, and embeddings) of Pythia-70M (Biderman et al., 2023). We closely follow Bricken et al. (2023), using a ReLU-linear encoder $f_i$ and sparse dimension $d_{\mathrm{SAE}} = 64 \times d$ and training the SAEs to minimize a combination of an L2 reconstruction loss and L1 regularization term which promotes sparsity. Details about our Pythia SAEs and their training can be found in Appendix B.1.
- The open source Gemma Scope SAEs (Lieberum et al., 2024) available for all sublayers (excluding embeddings) of the open-weights Gemma-2-2B model (Team et al., 2024). These SAEs use a Jump-ReLU-linear encoder and $d_{\mathrm{SAE}} = 8 \times d$. Details about the Gemma Scope SAEs can be found in Appendix B.2.

Scalably training better SAEs is an active area of research, as illustrated by the ready availability of open-source SAEs (Gao et al., 2024; Lieberum et al., 2024; Lin & Bloom, 2023). Thus, our goal is to—given a suite of trained SAEs—scalably apply them to understand NN behaviors; we treat scaling the SAEs themselves as out-of-scope.

**Attributing causal effects with linear approximations.** Let $m$ be a real-valued metric computed via a computational graph (e.g., a NN); let $\mathbf{a}$ represent a node in this graph. Following prior work (Vig et al., 2020; Finlayson et al., 2021), we quantify the importance of $\mathbf{a}$ on a pair of inputs $(x_{\mathrm{clean}}, x_{\mathrm{patch}})$ via its *indirect effect* (IE; Pearl, 2001) on $m$:

$$\mathrm{IE}(m; \mathbf{a}; x_{\mathrm{clean}}, x_{\mathrm{patch}}) = m\left(x_{\mathrm{clean}}|\mathrm{do}(\mathbf{a} = \mathbf{a}_{\mathrm{patch}})\right) - m(x_{\mathrm{clean}}). \tag{2}$$

Here, $\mathbf{a}_{\mathrm{patch}}$ is the value that $\mathbf{a}$ takes in the computation of $m(x_{\mathrm{patch}})$, and $m(x_{\mathrm{clean}}|\mathrm{do}(\mathbf{a} = \mathbf{a}_{\mathrm{patch}}))$ denotes the value of $m$ when computing $m(x_{\mathrm{clean}})$ but *intervening* in the computation of $m$ by manually setting $\mathbf{a}$ to $\mathbf{a}_{\mathrm{patch}}$. For example, given inputs $x_{\mathrm{clean}} =$"The **teacher**" and $x_{\mathrm{patch}} =$"The **teachers**," we have metric $m(x) = \log P(\text{"are"}|x) - \log P(\text{"is"}|x)$, the log probability difference output by the LM. Then if $\mathbf{a}$ is the activation of a particular neuron, a large value of $\mathrm{IE}(m; \mathbf{a}; x_{\mathrm{clean}}, x_{\mathrm{patch}})$ indicates that the neuron is highly influential on the model's decision to output "is" vs. "are" on this pair of inputs.

We often want to compute IEs for a very large number of model components $\mathbf{a} \in \mathbb{R}^d$, which cannot be done efficiently with (2). We thus employ linear approximations to (2) that can be computed for many $\mathbf{a}$ in parallel. The simplest such approximation, *attribution patching* (Nanda, 2022; Syed et al., 2023; Kramár et al., 2024), employs a first-order Taylor expansion

$$\hat{\mathrm{IE}}_{\mathrm{atp}}(m; \mathbf{a}; x_{\mathrm{clean}}, x_{\mathrm{patch}}) = \nabla_{\mathbf{a}} m|_{\mathbf{a} = \mathbf{a}_{\mathrm{clean}}} \left(\mathbf{a}_{\mathrm{patch}} - \mathbf{a}_{\mathrm{clean}}\right) \tag{3}$$

which estimates (2) for every $\mathbf{a}$ in parallel using only two forward and one backward pass.

To improve the quality of the approximation, we can instead employ a more expensive but more accurate approximation based on integrated gradients (Sundararajan et al., 2017; Hanna et al., 2024):

$$\hat{\mathrm{IE}}_{\mathrm{ig}}(m; \mathbf{a}; x_{\mathrm{clean}}, x_{\mathrm{patch}}) = \frac{1}{N} \left(\sum_{\alpha} \nabla_{\mathbf{a}} m|_{\alpha \mathbf{a}_{\mathrm{clean}} + (1-\alpha)\mathbf{a}_{\mathrm{patch}}}\right) \left(\mathbf{a}_{\mathrm{patch}} - \mathbf{a}_{\mathrm{clean}}\right) \tag{4}$$

where the sum in (4) ranges over $N = 10$ equally-spaced $\alpha \in \{0, \frac{1}{N}, \ldots, \frac{N-1}{N}\}$. This cannot be done in parallel for two nodes when one is downstream of another, but can be done in parallel for arbitrarily many nodes which do not depend on each other. Thus the additional cost of computing $\hat{\mathrm{IE}}_{\mathrm{ig}}$ over $\hat{\mathrm{IE}}_{\mathrm{atp}}$ scales linearly in $N$ and the serial depth of $m$'s computation graph.

The above discussion applies to the setting where we have a pair of clean and patch inputs, and we would like to understand that effect of patching a particular node from its clean to patch values. But in some settings (e.g., §4, 5), we have only a single input $x$. In this case, we instead use a *zero-ablation*, using the indirect effect $\mathrm{IE}(m; \mathbf{a}; x) = m(x|\mathrm{do}(\mathbf{a} = \mathbf{0})) - m(x)$ from setting $\mathbf{a}$ to $\mathbf{0}$. We get the modified formulas for $\hat{\mathrm{IE}}(m; \mathbf{a}; x)$ from (3) and (4) by replacing $\mathbf{a}$ with $\mathbf{0}$.

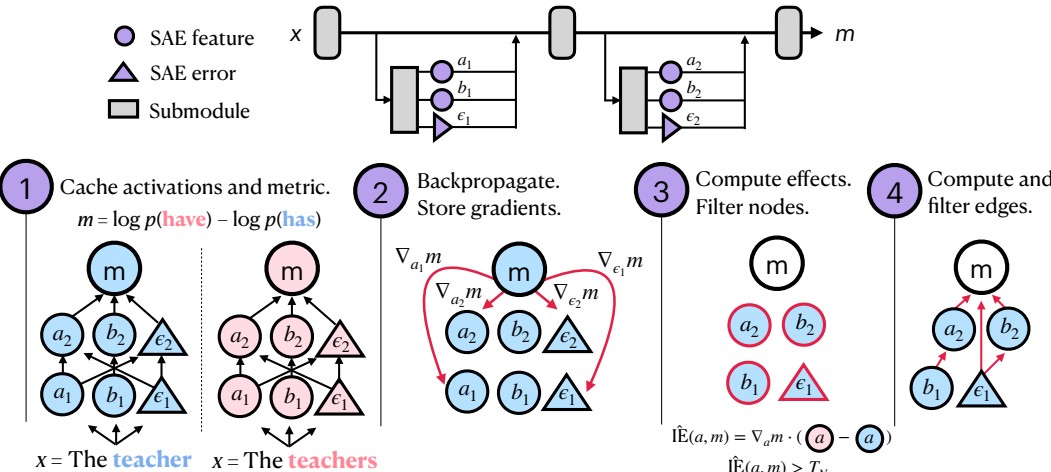

Figure 2: Overview of our method. We view our model as a computation graph that includes SAE features and errors. We cache activations (Step 1) and compute gradients (Step 2) for each node. We then compute approximate indirect effects with Eq. (3; shown) or (4) and filter according to a node threshold $T_N$ (Step 3). We similarly compute and filter edges (Step 4); see App. A.1.

## 3 SPARSE FEATURE CIRCUIT DISCOVERY

### 3.1 METHOD

Suppose we are given an LM $M$, SAEs for various submodules of $M$ (e.g., attention outputs, MLP outputs, and residual stream vectors, as in §2), a dataset $\mathcal{D}$ consisting either of contrastive pairs $(x_{\text{clean}}, x_{\text{patch}})$ of inputs or of single inputs $x$, and a metric $m$ that depends on $M$'s output when processing data from $\mathcal{D}$. For example, Figure 2 shows the case where $\mathcal{D}$ consists of pairs of inputs which differ in number, and $m$ is the log probability difference between $M$ outputting the verb form that is correct for the patch vs. clean input.

**Viewing SAE features as part of the model.** A key idea underpinning our method is that, by applying the decomposition (1) to various hidden states $\mathbf{x}$ in the LM, we can view the feature activations $f_i$ and SAE errors $\epsilon$ as being part of the LM's computation. We can thus represent the model as a computation graph $G$ where nodes correspond to feature activations or SAE errors at particular token positions.

**Approximating the IE of each node.** Let $\hat{\text{IE}}$ be one of $\hat{\text{IE}}_{\text{atp}}$ or $\hat{\text{IE}}_{\text{ig}}$ (see §2). Then for each node $\mathbf{a}$ in $G$ and input $x \sim \mathcal{D}$, we compute $\hat{\text{IE}}(m; \mathbf{a}; x)$; we then average over $x \sim \mathcal{D}$ to produce a score $\hat{\text{IE}}(m; \mathbf{a})$ and filter for nodes with $|\hat{\text{IE}}(m; \mathbf{a})| > T_N$ for some choice $T_N$ of node threshold.

Consistent with prior work (Nanda, 2022; Kramár et al., 2024), we find that $\hat{\text{IE}}_{\text{atp}}$ accurately estimates IEs for SAE features and SAE errors, with the exception of nodes in the layer 0 MLP and early residual stream layers, where $\hat{\text{IE}}_{\text{atp}}$ underestimates the true IE. We find that $\hat{\text{IE}}_{\text{ig}}$ significantly improves accuracy over $\hat{\text{IE}}_{\text{atp}}$ for these components, so we use it in our experiments below. See Appendix H for more information about linear approximation quality.

**Approximating the IE of edges.** Using an analogous linear approximation, we also compute the average IE of edges in the computation graph. Although the idea is simple, the mathematics are somewhat involved, so we relegate the details to App. A.1. After computing these IEs, we filter for edges with absolute IE exceeding some edge threshold $T_E$.

**Aggregation across token positions and examples.** For templatic data where tokens in matching positions play consistent roles (see §3.2, 3.3), we take the mean effect of nodes/edges across examples. For non-templatic data (§4, 5) we first sum the effects of corresponding nodes/edges across token position before taking the example-wise mean. See App. A.2.

| Structure | Example *clean* input | Example output |
|---|---|---|
| Simple | The **parents** | $p(\textbf{is}) - p(\textbf{are})$ |
| Within RC | The athlete that the **managers** | $p(\textbf{likes}) - p(\textbf{like})$ |
| Across RC | The **athlete** that the managers like | $p(\textbf{do}) - p(\textbf{does})$ |
| Across PP | The **secretaries** near the cars | $p(\textbf{has}) - p(\textbf{have})$ |

Table 1: Example clean inputs $x$ and outputs $m$ for subject-verb agreement tasks.

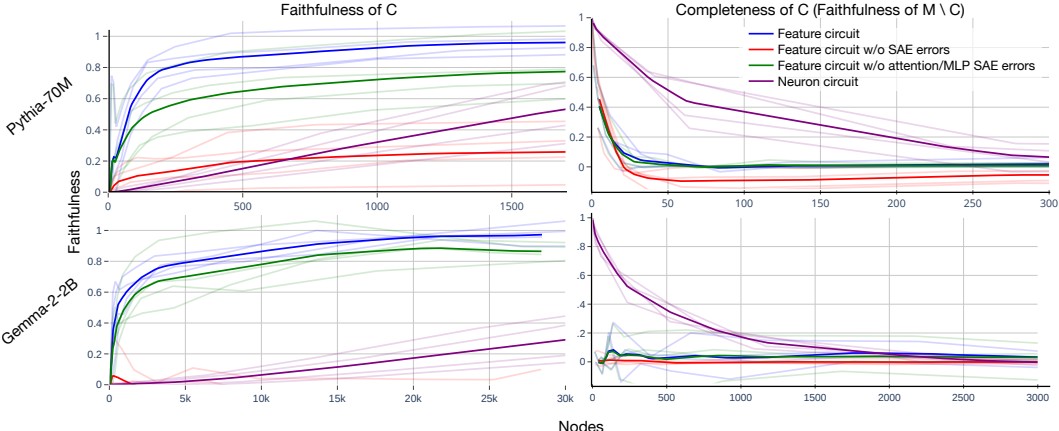

Figure 3: Faithfulness and completeness scores for circuits, measured on held-out data. Faint lines correspond to the structures from Table 1, with the average across structures in bold. The ideal faithfulness for circuits is 1, while the ideal completeness is 0.

**Practical considerations.** Various practical difficulties arise for efficiently computing the gradients needed by our method. We solve these using a combination of stop gradients, pass-through gradients, and tricks for efficient Jacobian-vector product computation; see App. A.3.

## 3.2 DISCOVERING AND EVALUATING SPARSE FEATURE CIRCUITS FOR SUBJECT-VERB AGREEMENT

To evaluate our method, we discover sparse feature circuits (henceforth, feature circuits) on Pythia-70M and Gemma-2-2B for four variants of the subject-verb agreement task (Table 1). Specifically, we adapt data from Finlayson et al. (2021) to produce datasets consisting of contrastive pairs of inputs that differ only in the grammatical number of the subject; the model's task is to choose the appropriate verb inflection.

We evaluate circuits for **interpretability**, **faithfulness**, and **completeness**. For each criterion, we compare to *neuron* circuits discovered by applying our methods with neurons in place of sparse features; in this setting, there are no error terms $\epsilon$. When evaluating feature circuits for faithfuless and completeness, we use a test split of our dataset, consisting of contrastive pairs not used to discover the circuit.

**Interpretability.** For Pythia SAEs, we asked human crowdworkers to rate the interpretability of random features, random neurons, features from our feature circuits, and neurons from our neuron circuits. Crowdworkers rated sparse features as significantly more interpretable than neurons, with features that participate in our circuits also being more interpretable than randomly sampled ones (App. F). This replicates prior findings that SAE features are substantially more interpretable than neurons (Bricken et al., 2023). For Gemma-2 SAEs, we refer the reader to Lieberum et al. (2024), which finds the interpretability of these SAEs' features to be on par with those trained via other state-of-the-art techniques.

**Faithfulness.** Given a circuit $C$ and metric $m$, let $m(C)$ denote the average value of $m$ over inputs from $\mathcal{D}$ when running our model with all nodes outside of $C$ mean-ablated, i.e., set to their average

value over data from $\mathcal{D}$.[2] We then measure the faithfulness of a circuit as $\frac{m(C)-m(\varnothing)}{m(M)-m(\varnothing)}$, where $\varnothing$ denotes the empty circuit and $M$ denotes the full model. Intuitively, this metric captures the proportion of the model's performance our circuit explains, relative to mean ablating the full model (which represents the "prior" performance of the model when it is given information about the task, but not about specific inputs).

We find that components in early model layers are typically involved in processing specific tokens. In practice, the inputs in the train split of our dataset (used to discover the circuit) and the test split (for evaluation) do not contain identical tokens, making it difficult to evaluate the quality of early segments of our circuit. Thus, we ignore the first $1/3$ of our circuit, and only evaluate the latter $2/3$.

We plot faithfulness for feature circuits and neuron circuits after sweeping over node thresholds $T_N$ (Fig. 3). We find that small feature circuits explain a large proportion of model behavior: the majority of performance in Pythia-70M, resp. Gemma-2-2B is explained by only 100, resp. 500 nodes. In contrast, around 1500, resp. 50000 neurons are required to explain half the performance. However, as SAE error nodes are high-dimensional and coarse-grained, they cannot be fairly compared to neurons; we thus also plot the faithfulness of feature circuits with all SAE error nodes removed, or with all attention and MLP error nodes removed. Unsurprisingly, we find that removing residual stream SAE error nodes severely disrupts the model and curtails its maximum performance; removing MLP and attention error nodes is less disruptive.

**Completeness.** Are there parts of the model behavior that our circuit fails to capture? We measure this as the faithfulness of the circuit's complement $M \setminus C$ (Fig. 3). We observe that we can eliminate the model's task performance by ablating only a few nodes from our feature circuits, and that this is true even when we leave all SAE errors in place. In contrast, it takes hundreds (for Pythia) or thousands (for Gemma) of neurons to achieve the same effect.

### 3.3 Case study: Subject-verb agreement across a relative clause

We find that inspecting small feature circuits produced by our technique can provide insights into how Pythia-70M and Gemma-2-2B arrive at observed behaviors. To illustrate this, we present a case study of relatively small feature circuits for subject-verb agreement across a relative clause (RC).

To keep the number of nodes we need to annotate manageable, we tune our node threshold to produce a small circuit with faithfulness $> 0.2$. For Pythia, this results in a circuit with 86 nodes and faithfulness 0.21; for Gemma we study a circuit with 223 nodes and faithfulness 0.21. We summarize these circuits in Figure 4; the full circuits (as well as small circuits for other subject-verb agreement tasks) can be found in App. C.1. We depict SAE features with rectangles and SAE errors with triangles.

Our circuits depict interpretable algorithms wherein both models of study select appropriate verb forms via two pathways. The first pathway consists of features which detect the number of the main subject and then generically promote matching verb forms. The second pathway begins the same, but moves the relevant number information to the end of the relative clause by using PP/RC boundary detectors. Gemma 2 also uses noun phrase (NP) number trackers, which detect the number of the noun that heads an NP and remain active on all tokens until the end of the NP; these promote matching verb forms at each position, but especially at the last token of an NP.

We find significant overlap between this circuit and the circuit we discovered for agreement across a prepositional phrase, with Pythia-70M and Gemma-2-2B handling these syntactically distinct structures in a mostly uniform way. In accordance with Finlayson et al. (2021), we find less overlap with our circuits for simple agreement and within RC agreement (Appendix C.1).

## 4 Application: Removing unintended signals from a classifier without disambiguating labels

NN classifiers often rely on unintended signals—e.g., spurious features. Nearly all prior work on mitigating this problem relies on access to *disambiguating labeled data* in which unintended signals are less predictive of labels than intended ones. However, some tasks have structural properties

---

[2]Following Wang et al. (2023), we ablate features by setting them to their mean *position-specific* values.

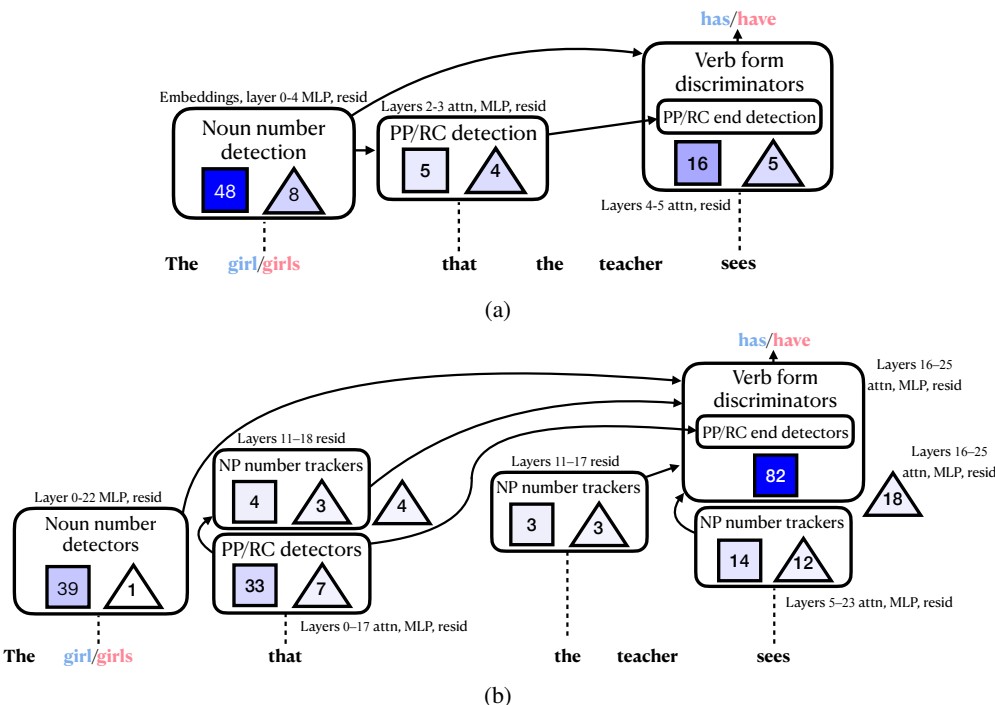

Figure 4: Summary of Pythia's (a) and Gemma 2's (b) circuits for agreement across RC (full circuits in App. C.1). The models detect the number of the subject. Then, they detect the start of a PP/RC modifying the subject. Verb form discriminators promote particular verb inflections (singular or plural). Gemma 2 additionally uses separate features to track the number of the noun that heads the current noun phrase. Squares show number of feature nodes in the group and triangles show number of SAE error nodes, with the shading indicating the sum of I͂E terms across nodes in the group. As we cannot directly interpret the triangles, we rely on their positions or inclusion in other groups to label them. If the label is ambiguous, we leave the triangles outside the boxes.

which disallow this assumption. For example, inputs for different classes might come from different data sources (Zech et al., 2018). Additionally, some have raised concerns (Ngo et al., 2024; Casper et al., 2023) that sophisticated LMs trained with human feedback (Christiano et al., 2023) in settings with easy-to-hard domain shift (Burns et al., 2023; Hase et al., 2024) will be misaligned because, in these settings, "overseer approval" and "desirable behavior" are equally predictive of training reward labels. More fundamentally, the problem with unintended signals is that they are *unintended*—not they are insufficiently predictive—and we would like our methods to reflect this.

We thus propose Spurious Human-interpretable Feature Trimming (SHIFT), where a human inspects a classifier's feature circuit and removes features which they judge to be task-irrelevant. We show that SHIFT removes sensitivity to unintended signals without access to disambiguating labeled data, or even without knowing what the signals are ahead of time.

**Method.** Suppose we are given labeled training data $\mathcal{D} = \{(x_i, y_i)\}$; an LM-based classifier $C$ trained on $\mathcal{D}$; and SAEs for various components of $C$. To perform SHIFT, we:

1. Apply the methods from §3 to compute a feature circuit that explains $C$'s accuracy on inputs $(x, y) \sim \mathcal{D}$ (e.g., using metric $m = -\log C(y|x)$).
2. Manually inspect and evaluate for task-relevancy each feature in the circuit from Step 1.
3. Ablate from $C$ features judged to be task-irrelevant to obtain a classifier $C'$.
4. (Optional) Further fine-tune $C'$ on data from $\mathcal{D}$.

Step 3 removes the classifier's dependence on unintended signals we can identify, but may disrupt the classifier's performance for the intended signal. Step 4 can be used to restore some performance.

| Method | Pythia-70M | | | Gemma-2-2B | | |
|---|---|---|---|---|---|---|
| | ↑Profession | ↓Gender | ↑Worst group | ↑Profession | ↓Gender | ↑Worst group |
| Original | 61.9 | 87.4 | 24.4 | 67.7 | 81.9 | 18.2 |
| CBP | 83.3 | 60.1 | 67.7 | 90.2 | **50.1** | 86.7 |
| Random | 61.8 | 87.5 | 24.4 | 67.3 | 82.3 | 18.0 |
| SHIFT | 88.5 | 54.0 | 76.0 | 76.0 | 51.5 | 50.0 |
| SHIFT + retrain | **93.1** | **52.0** | **89.0** | **95.0** | 52.4 | **92.9** |
| Neuron skyline | 75.5 | 73.2 | 41.5 | 65.1 | 84.3 | 5.6 |
| Feature skyline | 88.5 | 54.3 | 62.9 | 80.8 | 53.7 | 56.7 |
| Oracle | 93.0 | 49.4 | 91.9 | 95.0 | 50.6 | 93.1 |

Table 2: Accuracies on balanced data for the intended label (profession) and unintended label (gender). "Worst group accuracy" refers to whichever profession accuracy is lowest among male professors, male nurses, female professors, female nurses.

**Experimental setup.** We illustrate SHIFT using the Bias in Bios dataset (BiB; De-Arteaga et al., 2019). BiB consists of professional biographies, and the task is to classify an individual's profession based on their biography. BiB also provides labels for a spurious feature: gender. We subsample BiB to produce two sets of labeled data:

- The **ambiguous set**, consisting of bios of male professors (labeled 0) and female nurses (labeled 1).
- The **balanced set**, consisting of an equal number of bios for male professors, male nurses, female professors, and female nurses. These data carry **profession labels** (the intended signal) and **gender labels** (the unintended signal).

The **ambiguous set** represents a worst-case scenario: the unintended signal is perfectly predictive of training labels. Given only access to the **ambiguous set**, our task is to produce a **profession** classifier which is accurate on the **balanced set**.

We adapt Pythia-70M and Gemma-2-2B into classifiers by training linear classification heads with the **ambiguous set**; see App. E.1 for probe training details. We then discover feature circuits for these classifiers using the zero-ablation variant described in §3.1; the Pythia circuit contains 67 features, and the Gemma circuit contains 46. We manually interpret each feature using the Neuronpedia interface (Lin & Bloom, 2023), which displays maximally activating dataset exemplars on a large text corpus, as the features' direct effects on output logits. We judge 55 of the Pythia features and 43 of the Gemma features to be task-irrelevant—e.g., features that promote female-associated language in biographies of women, as in Figure 19 (see App. D for more examples features). Although this interpretability step uses additional unlabeled data, we emphasize that we never use additional *labeled* data (or even additional unlabeled classification data).

To apply SHIFT, we zero-ablate these irrelevant features. Finally, we retrain the linear classification head with the **ambiguous set** using activations extracted from the ablated model. We evaluate all accuracies on the **balanced set**.

**Baselines and skylines.** To contextualize the performance of SHIFT, we also implement:

- **SHIFT with neurons.** Perform SHIFT, but using neurons instead of SAE features.
- **Concept Bottleneck Probing** (CBP), adapted from Yan et al. (2023) (originally for multimodal text/image models). CBP works by training a probe to classify inputs $x$ given access only to a vector of affinities between the LM's representation of $x$ and various concept vectors. See App. E.2 for implementation details.
- **Random feature ablations.** Perform SHIFT, but using (the same number of) randomly selected SAE features instead of features selected by a human annotator.
- **Feature skyline.** Instead of relying on human judgement to evaluate whether a feature should be ablated, we zero-ablate the 55 (for Pythia) or 43 (for Gemma) features from our circuit that are most causally implicated in **spurious feature** accuracy on the **balanced set**.
- **Neuron skyline.** The same as the feature skyline, but mean-ablating 55 or 43 neurons.
- **Oracle.** A classifier trained on **ground-truth labels** on the **balanced set**.

**Results.** We find (Table 2) that SHIFT almost completely removes the classifiers' dependence on gender information for both models. In the case of Gemma (but not Pythia), the feature ablations

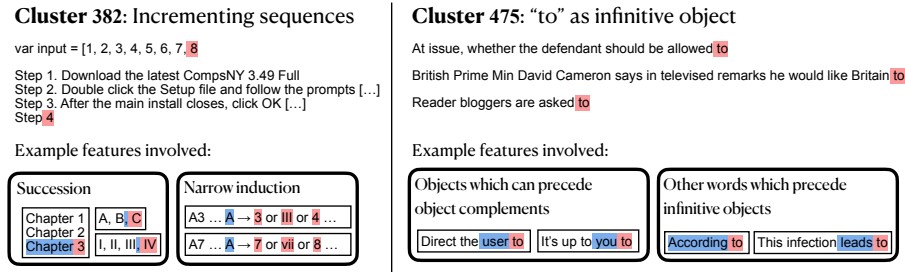

Figure 5: Example clusters and features which participate in their circuits (see App. C.3 for the full circuits). Features are active on tokens shaded in blue and promote tokens shaded in red. *(left)* An example *narrow* induction feature recognizes the pattern $A3 \ldots A$ and copies information from the 3 token. This composes with a succession feature to implement the prediction $A3 \ldots A \rightarrow 4$. *(right)* One feature promotes "to" after words which can take infinitive objects. A separate feature activates on objects of verbs or prepositions and promotes "to" as an object complement.

damage model performance; however, this performance is restored (without reintroducing the bias) by further training on the ambiguous set. Comparing SHIFT without retraining to the feature skyline, we further observe that SHIFT optimally or near-optimally identifies the best features to remove.

SHIFT critically relies on the use of properly selected SAE features. When ablating random SAE features, we see essentially no effect on probe performance. When applying SHIFT with neurons, essentially none of the neurons are interpretable, making it difficult to tell if they ought to be ablated; see Appendix D for examples. Because of this, we abandon the SHIFT with neurons baseline. Even using the **balanced set** to automatically select neurons for removal (the neuron skyline) fails to match SHIFT's performance, as the neurons most implicated in **spurious feature** classification are also useful for **ground-truth** classification.

## 5 UNSUPERVISED CIRCUIT DISCOVERY AT SCALE

Previous work on circuit analysis relied on human-collected datasets to specify LM behaviors (Wang et al., 2023; Conmy et al., 2023; Hanna et al., 2023). However, LMs implement numerous interesting behaviors, many of which may be counterintuitive to humans. In this section, we adapt our techniques to produce a near-fully-automated interpretability pipeline, starting from a large text corpus—here, a large subset of The Pile (Gao et al., 2020)—and ending with thousands of feature circuits for auto-discovered model behaviors. These experiments are performed with Pythia-70M.

We proceed in two steps:

1. **Behavior discovery via clustering.** We interpret our large text corpus as a dataset $\{(x_i, y_i)\}$ of contexts $x_i$ with ground-truth next tokens $y_i$. Following Michaud et al. (2023), we associate a vector $\mathbf{v}_i = \mathbf{v}(x_i, y_i)$ to each sample and apply a clustering algorithm to $\{\mathbf{v}_i\}$; this segments our large corpus into a number of smaller subcorpora corresponding to the clusters. Although this approach is entirely unsupervised, many of the resulting subcorpora capture human-interpretable model behaviors, such as predicting the next number in a sequence (Fig 5). We experiment with a number of ways of assigning $(x_i, y_i) \mapsto \mathbf{v}_i$, such as using the training gradient $\nabla_\theta \log P_\theta(y_i|x_i)$ as in Michaud et al. (2023) as well as approaches which leverage SAE activations or gradients. See App. G for details.

2. **Circuit discovery.** Given a subcorpus $\mathcal{D} = \{(x_i, y_i)\}$, we apply the zero-ablation variant of our feature circuit discovery technique from §3 using the dataset $\mathcal{D}$ and metric $m = -\log P(y_i|x_i)$. Thus, to each subcorpus we associate a feature circuit.

We present example clusters, as well as interesting features participating in their associated circuits (Figure 5). An interface for exploring all of our clusters and (unlabeled) circuits can be found at `feature-circuits.xyz`.

While evaluating these clusters and circuits is an important open problem, we generally find that these clusters expose interesting LM behaviors, and that their respective feature circuits can provide

useful insights on mechanisms of LM behavior. For instance, we automatically discover attention features implicated in succession and induction, two phenomena thoroughly studied in prior work at the attention head level using human-curated data (Olsson et al., 2022; Gould et al., 2023).

Feature circuits can also shed interesting light on their clusters. For example, while the clusters in Figure 5 seem at first to each represent a single mechanism, circuit-level analysis reveals in both cases a union of distinct mechanisms. For cluster 475, Pythia-70M determines whether "to [verb]" is an appropriate object in two distinct manners (see Figure 5 caption). And for cluster 382, the prediction of successors relies on general succession features, as well as multiple *narrow* induction features which recognize patterns like "$A3 \ldots A$".

## 6 RELATED WORK

**Causal interpretability.** Interpretability research has applied causal mediation analysis (Pearl, 2001; Robins & Greenland, 1992) to understand the mechanisms underlying particular model behaviors and their emergence (Yu et al., 2023; Geva et al., 2023; Hanna et al., 2023; Todd et al., 2024; Prakash et al., 2024; Chen et al., 2024, *inter alia*). This typically relies on counterfactual interventions (Lewis, 1973), such as activation patching or path patching on coarse-grained components (Conmy et al., 2023; Wang et al., 2023). Some techniques aim to, given a hypothesized causal graph, identify a matching causal mechanism in an LM (Geiger et al., 2021; 2022; 2023); in contrast, we aim here to discover causal mechansisms without starting from such hypotheses.

**Robustness to spurious correlations.** There is a large literature on mitigating robustness to spurious correlations, including techniques which rely on directly optimizing worst-group accuracy (Sagawa et al., 2020; Oren et al., 2019; Zhang et al., 2021; Sohoni et al., 2022; Nam et al., 2022), automatically or manually reweighting data between groups (Liu et al., 2021; Nam et al., 2020; Yaghoobzadeh et al., 2021; Utama et al., 2020; Creager et al., 2021; Idrissi et al., 2022; Orgad & Belinkov, 2023), training classifiers with more favorable inductive biases (Kirichenko et al., 2023; Zhang et al., 2022; Iskander et al., 2024), or editing out undesired concepts (Iskander et al., 2023; Belrose et al., 2023; Wang et al., 2020; Ravfogel et al., 2020; 2022a;b). All of these techniques rely on access to disambiguating labeled data in the sense of §4. Some techniques from a smaller literature focused on image or multimodal models apply without such data (Oikarinen et al., 2023; Yan et al., 2023). Our method here is inspired by the approach of Gandelsman et al. (2024) based on interpreting and ablating undesired attention heads in CLIP.

**Feature disentanglement.** In addition to recent work on SAEs for LM interpretability (Cunningham et al., 2024; Bricken et al., 2023; Gao et al., 2024; Rajamanoharan et al., 2024a;b), other approaches to feature disentanglement include Schmidhuber (1992); Desjardins et al. (2012); Kim & Mnih (2018); Chen et al. (2016); Makhzani & Frey (2013); He et al. (2022); Peebles et al. (2020); Schneider & Vlachos (2021); Burgess et al. (2017); Chen et al. (2018); Higgins et al. (2017); *i.a.*

## 7 CONCLUSION

We have introduced a method for discovering circuits on sparse features. Using this method, we discover human-interpretable causal graphs for a subject-verb agreement task, a classifier, and thousands of general token prediction tasks. We can edit the set of features that models have access to by ablating sparse features that humans deem spurious; we find that this is significantly more effective than a neuron-based ablation method which has an unfair advantage.

## 8 LIMITATIONS

The success of our technique relies on access to SAEs for a given model. Training such SAEs currently requires a large (but one-time) upfront compute cost. Additionally, model components not captured by the SAEs will remain uninterpretable after applying our method.

Much of our evaluation is qualitative. While we have quantitative evidence that feature circuits are useful for improving generalization without additional data (§4), evaluating dictionaries and circuits without downstream tasks is challenging. Feature labeling is also a qualitative process; thus, labels may vary across annotators, and may vary depending on the task of interest.

## REPRODUCIBILITY

We release code, data and autoencoders at `github.com/saprmarks/feature-circuits`. Experimental details can be found in appendices A, E, and G. Our experiments are conducted entirely on open-weights models. The Gemma Scope SAEs are publicly available (Lieberum et al., 2024). Features for both SAE suites can be browsed on Neuronpedia (Lin & Bloom, 2023). Our clusters and associated feature circuits can be browsed at `feature-circuits.xyz`.

## ACKNOWLEDGMENTS

We thank Stephen Casper, Buck Schlegeris, Ryan Greenblatt, and Neel Nanda for discussion of ideas upstream to the experiments in §4. We thank Logan Riggs and Jannik Brinkmann for help training SAEs. We also thank Josh Engels and Max Tegmark for discussions about clustering and sparse projections related to §5. S.M. is supported by an Open Philanthropy alignment grant. C.R. is supported by Manifund Regrants. E.J.M. is supported by the NSF via the Graduate Research Fellowship Program (Grant No. 2141064) and under Cooperative Agreement PHY-2019786 (IAIFI). Y.B. is supported by the Israel Science Foundation (Grant No. 448/20) and an Azrieli Foundation Early Career Faculty Fellowship. Y.B. and D.B. are supported by a joint Open Philanthropy alignment grant. A.M. is supported by a Zuckerman postdoctoral fellowship.

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

## A    METHODOLOGICAL DETAILS FOR FEATURE CIRCUIT DISCOVERY

### A.1    COMPUTING EDGE WEIGHTS

Let $e$ be an edge between an upstream node $\mathbf{u}$ and downstream node $\mathbf{d}$; let also $\mathcal{M}$ be the set of nodes $\mathbf{m}$ intermediate between $\mathbf{u}$ and $\mathbf{d}$. We define the weight of the edge $e$ to be the effect on the

Figure 6: Aggregation of node/edge effects across examples (and sometimes, across token positions). Each feature is labeled as "token position, feature index." If we have templatic data, we preserve token position information, and treat the same features in different token positions as different features. If we have more general non-templatic data, we first sum across positions, and then take the example-wise mean of the position-aggregated effects.

metric $m$ when intervening to set

$$\mathbf{d} = \mathbf{d}\left(x_{\text{clean}} | \text{do}\left(\mathbf{u} = \mathbf{u}_{\text{patch}}, \mathbf{m} = \mathbf{m}_{\text{clean}} : \mathbf{m} \in \mathcal{M}\right)\right).$$

Intuitively, this captures the indirect effect of $\mathbf{u}$ on $m$ via the *direct* effect on $\mathbf{d}$, but excluding effects on $\mathbf{d}$ mediated by some other intermediate node $\mathbf{m}$.

As with nodes, we employ a linear approximation:

$$\hat{\text{IE}}(m; e; x_{\text{clean}}, x_{\text{patch}}) = \nabla_{\mathbf{d}} m\big|_{\mathbf{d}_{\text{clean}}} \nabla_{\mathbf{u},\text{stop}(\mathcal{M})} \mathbf{d}\big|_{\mathbf{u}_{\text{clean}}} \left(\mathbf{u}_{\text{patch}} - \mathbf{u}_{\text{clean}}\right) \tag{5}$$

where $\nabla_{\mathbf{u},\text{stop}(\mathcal{M})} \mathbf{d}$ denotes the gradient of $\mathbf{d}$ with respect to $\mathbf{u}$ when treating all $\mathbf{m} \in \mathcal{M}$ as constant. In practice, this is computed by applying stop-gradients to all intermediate nodes $\mathbf{m}$ during pytorch's backwards pass.

If $\mathbf{d}$ is an SAE error, then the naive approach to computing is expression involves performing $d_{\text{model}}$ backwards passes; fortunately we can still compute the product in a single backwards pass as explained in §A.3.

## A.2 AGGREGATING ACROSS TOKEN POSITIONS AND EXAMPLES

Figure 6 summarizes how we aggregate effects across examples (and optionally across token positions). For templatic data where tokens in matching positions play consistent roles (see §3.2, 3.3), we take the mean effect of nodes/edges across examples. In this case, we treat the same feature (or neuron) in different token positions as different nodes altogether in the circuit, each with their own separate effects on target metric $m$.

For non-templatic data (§4, 5), we first sum the effects of corresponding nodes/edges across token positions before taking the example-wise mean. This means that each feature appears in the circuit once, representing its effects at all token positions in an input.

## A.3 PRACTICAL CONSIDERATIONS

Here we review a number of tricks that we use to compute the quantities defined above efficiently. The backbone of our approach is to, given an activation $\mathbf{x} \in \mathbb{R}^{d_{\text{model}}}$ of some submodule for which we have an SAE, use the SAE to compute the quantities $f_i(\mathbf{x})$ and $\boldsymbol{\epsilon}(\mathbf{x})$ in (1), and then intervene in our model's forward pass to set

$$\mathbf{x} \leftarrow \sum_i f_i(\mathbf{x})\mathbf{v}_i + \mathbf{b} + \boldsymbol{\epsilon}(\mathbf{x}). \tag{6}$$

Even though $\mathbf{x}$ was already numerically equal to the right-hand side of (6), after the intervention the computation graph will incorporate the variables $f_i(\mathbf{x})$ and $\boldsymbol{\epsilon}(\mathbf{x})$. Thus, when we use Pytorch's

autograd algorithm to peform backpropogation of downstream quantities, we will automatically compute gradients for these variables.

An alternative approach for computing gradients (which we do not use) is to simply run the model without interventions, use backpropogation to compute all gradients $\nabla_{\mathbf{x}} m$, and use the formulas

$$\nabla_{f_i} m = \nabla_{\mathbf{x}} m \cdot \mathbf{v}_i, \qquad \nabla_{\boldsymbol{\epsilon}} m = \nabla_{\mathbf{x}} m$$

which follow from the chain rule when $m$ is any function of $\mathbf{x}$.

**Stop gradients on SAE errors to compute SAE feature gradients.** The natural way to compute the SAE error $\boldsymbol{\epsilon}(\mathbf{x})$ is by first using the SAE to compute $\hat{\mathbf{x}}$ and then setting $\boldsymbol{\epsilon}(\mathbf{x}) = \mathbf{x} - \hat{\mathbf{x}}$. However, if we take this approach, then after applying the intervention (6) we would have

$$\nabla_{f_i} m = \nabla_{vx^d} m \nabla_{f_i} \mathbf{x}^d = \nabla_{\mathbf{x}^d} m \nabla_{f_i} (\hat{\mathbf{x}} + \mathbf{x}^u - \hat{\mathbf{x}}) = 0$$

where $\mathbf{x}^d$ is the copy of $\mathbf{x}$ downstream of $f_i$ in the computation graph, and $\mathbf{x}^u$ is the copy upstream of $f_i$. To fix this, we apply a stop gradient to $\boldsymbol{\epsilon}(\mathbf{x})$ so that $\mathbf{x}^d = \hat{\mathbf{x}} + \text{stopgrad}(\mathbf{x}^u - \hat{\mathbf{x}})$.

**Pass-through gradients.** Although the stop gradient from above solves the problem of vanishing gradients for the $f_i$, it interferes with the backpropogation of gradients to further upstream nodes. In order to restore exact gradient computation, we implement a pass-through gradient on the computation of our dictionary. That is, in the notation above, we intervene in the *backwards* pass of our model to set

$$\nabla_{\mathbf{x}^u} m \leftarrow \nabla_{\mathbf{x}^d} m.$$

**Jacobian-vector products.** Done naively, computing the quantity in (5) when $\mathbf{d}$ is an SAE errors would take $O(d_{\text{model}})$ backwards passes. Fortunately, one can use the following trick: when $A$ is a constant $1 \times n$ matrix, $\mathbf{x} \in \mathbb{R}^m$, and $\mathbf{y} = \mathbf{y}(\mathbf{x}) \in \mathbb{R}^n$ is a function of $\mathbf{x}$, we have

$$A \nabla_{\mathbf{x}} \mathbf{y} = \nabla_{\mathbf{x}} (A \mathbf{y})$$

where the right-hand side is a $1 \times m$ Jacobian which can be computed with a single backward pass. Thus we can compute (5) with only two backwards passes by first computing $\nabla_{\mathbf{d}} m |_{\mathbf{d}_{\text{clean}}}$ and then computing $\nabla_{\mathbf{u}} \left( \nabla_{\mathbf{d}} m |_{\mathbf{d}_{\text{clean}}} \right)$ with another backwards pass, where the second $\nabla_{\mathbf{d}} m |_{\mathbf{d}_{\text{clean}}}$ is treated as a constant (e.g., by detaching it in Pytorch).

# B DETAILS ON SPARSE AUTOENCODERS

## B.1 PYTHIA-70M SPARSE AUTOENCODERS

### B.1.1 ARCHITECTURE

Following Bricken et al. (2023), our SAEs for Pythia-70M are one-layer MLPs with a tied pre-encoder bias. In more detail, our SAEs have parameters

$$W_E \in \mathbb{R}^{d_{\text{SAE}} \times d_{\text{model}}}, W_D \in \mathbb{R}^{d_{\text{model}} \times d_{\text{SAE}}}, \qquad b_E \in \mathbb{R}^{d_{\text{SAE}}}, b_D \in \mathbb{R}^{d_{\text{model}}}$$

where the columns of $W_D$ are constrained to be unit vectors. Given an input activation $\mathbf{x} \in \mathbb{R}^{d_{\text{model}}}$, we compute the sparse features activations via

$$\mathbf{f} = [f_1(\mathbf{x}) \quad \dots \quad f_{d_{\text{SAE}}}(\mathbf{x})] = \text{ReLU}(W_E(\mathbf{x} - \mathbf{b}_D) + \mathbf{b}_E)$$

with the ReLU nonlinearity applied coordinatewise and reconstructions via

$$\hat{\mathbf{x}} = W_D \mathbf{f} + \mathbf{b}_D.$$

The feature vectors $\mathbf{v}_i \in \mathbb{R}^{d_{\text{model}}}$ are the columns of $W_D$.

### B.1.2 TRAINING

Fix a specific choice of activation in Pythia-70M, e.g. MLP output, attention output, or residual stream in a particular layer. Following Cunningham et al. (2024); Bricken et al. (2023) we train an SAE for this activation by sampling random text from The Pile (Gao et al., 2020) (specifically the first 128 tokens of random documents), extracting the values $\mathbf{x}$ for this activation over every token, and then training our SAE to minimize a loss function

$$\mathcal{L} = \mathcal{L}_{\text{reconstruction}} + \lambda \mathcal{L}_{\text{sparsity}} = \|\hat{\mathbf{x}} - \mathbf{x}\|_2 + \lambda \|\mathbf{f}\|_1$$

consisting of a L2 reconstruction loss and a L1 regularization term to promote sparsity. This loss is optimized using a variant of Adam (Kingma & Ba, 2014) adapted to ensure that the columns of $W_D$ are unit vectors (see Bricken et al. (2023) or our code for details). We use $\lambda = 0.1$ and a learning rate of $10^{-4}$.

Following Nanda (2023), we cache activations from 10000 contexts in a buffer and randomly sample batches of size $2^{14}$ for training our SAE. When the buffer is half-depleted, we replenish it with fresh tokens from The Pile. We train for 120000 steps, resulting in a total of about 2 billion training tokens.

A major obstacle in training SAEs is *dead features*, that is, neurons in the middle layer of the SAE which never or rarely activate. We mitigate this by, every 25000 training steps, reinitializing features which have not activated in the previous 12500 steps using the same reinitialization procedure described in Bricken et al. (2023).

Finally, we use a linear learning rate warmup of 1000 steps at the start of training and after every time that neurons are resampled.

### B.1.3 EVALUATION

Here we report on various easy-to-quantify metrics of SAE quality. Note that these metrics leave out important qualitative properties of these SAEs, such as the interpretability of their features (App. F). Our metrics are:

- **Variance explained**, as measured by $1 - \frac{\text{Var}(\mathbf{x} - \hat{\mathbf{x}})}{\text{Var}(\mathbf{x})}$.
- Average **L1**, and **L0** norms of $\mathbf{f}$.
- **Percentage of features alive** as measured by features which activate at least once on a batch of $512$ tokens.
- **Cross entropy** (CE) **difference** and **percentage of CE recovered**. The CE difference is the difference between the model's original CE loss and the model's CE loss when intervening to set $\mathbf{x}$ to the reconstruction $\hat{\mathbf{x}}$. We obtain percentage of CE recovered by dividing this difference by the difference between the original CE loss and the CE loss when zero-ablating $\mathbf{x}$. These CE losses are computed averaged over a batch of $128$ contexts of length $128$.

These metrics are shown in Tables 3–6. Note that we index residual stream activations to be the layer which *outputs* the activation (so the layer 0 residual stream is *not* the embeddings, and the layer 5 residual stream is the output of the final layer, immediately preceding the final decoder).

| % Variance Explained | L1 | L0 | % Alive | CE Diff | % CE Recovered |
|---|---|---|---|---|---|
| 96 | 1 | 3 | 36 | 0.17 | 98 |

Table 3: Embedding SAE evaluation.

### B.2 GEMMA-2-2B SPARSE AUTOENCODERS

For Gemma-2-2B, we use the Gemma Scope SAEs released by Lieberum et al. (2024), which are based on the Jump-ReLU architecture proposed by Rajamanoharan et al. (2024b). We use the SAEs of width 16384. There exist SAEs for the attention, MLP, and residual vectors for each of the 26

| Layer | % Variance Explained | L1 | L0 | % Alive | CE Diff | % CE Recovered |
|-------|---------------------|----|----|---------|---------|----------------|
| Attn 0 | 92% | 8 | 128 | 17% | 0.02 | 99% |
| Attn 1 | 87% | 9 | 127 | 17% | 0.03 | 94% |
| Attn 2 | 90% | 19 | 215 | 12% | 0.05 | 93% |
| Attn 3 | 89% | 12 | 169 | 13% | 0.03 | 93% |
| Attn 4 | 83% | 8 | 132 | 14% | 0.01 | 95% |
| Attn 5 | 89% | 11 | 144 | 20% | 0.02 | 93% |

Table 4: Attention SAE evaluation by layer.

| Layer | % Variance Explained | L1 | L0 | % Alive | CE Diff | % CE Recovered |
|-------|---------------------|----|----|---------|---------|----------------|
| MLP 0 | 97% | 5 | 5 | 40% | 0.10 | 99% |
| MLP 1 | 85% | 8 | 69 | 44% | 0.06 | 95% |
| MLP 2 | 99% | 12 | 88 | 31% | 0.11 | 88% |
| MLP 3 | 88% | 20 | 160 | 25% | 0.12 | 94% |
| MLP 4 | 92% | 20 | 100 | 29% | 0.14 | 90% |
| MLP 5 | 96% | 31 | 102 | 35% | 0.15 | 97% |

Table 5: MLP SAE evaluation by layer.

layers in the model. However, the attention and MLP SAEs are trained at different positions than in Pythia: attention SAEs are trained on the *input* to the out projection, and MLP SAEs are trained on the output of the LayerNorm following the MLP. The embedding SAEs are experimental and have a dictionary size of only 4000, so we do not use them in our experiments.

There exist multiple SAEs for every submodule. The primary difference between them is their average L0 norm.[3] Neuronpedia (Lin & Bloom, 2023) uses the SAEs with the L0 norm closest to 100; we do the same.

One of the primary technical challenges in using the Gemma Scope SAEs is the existence of BOS features. These are features that are active primarily or only on BOS tokens, and whose top logits are generally not informative. These features are difficult to interpret, but can have high indirect effects on the model's logits. As we cannot interpret them, we exclude them from annotation and from the SHIFT analysis (i.e., we do not ablate them). We also exclude them when running the feature skyline in §4.

## C  FEATURE CIRCUITS

### C.1  SUBJECT-VERB AGREEMENT

Here, we present the full agreement circuits for various syntactic agreement structures, with researcher-provided annotations for features. We chose thresholds manually in order to keep the number of nodes to annotate manageable while still displaying the full range of feature types for a given task.

In each circuit, sparse features are shown in rectangles, whereas causally relevant error terms not yet captured by our SAEs are shown in triangles. Nodes shaded in darker colors have stronger effects on the target metric $m$. Blue nodes and edges are those which have positive indirect effects (i.e., are useful for performing the task correctly), whereas red nodes and edges are those which have counterproductive effects on $m$ (i.e., cause the model to consistently predict incorrect answers).

First, we present agreement across a relative clause. Pythia (Figure 7) and Gemma (Figure 8) both appear to detect the subject's grammatical number at the subject position. One position later, features detect the presence of relative pronouns (the start of the distractor clause). Finally, at the last token of the relative clause, the attention moves the subject information to the last position, where it assists in predicting the correct verb inflection. Gemma 2 additionally leverages noun phrase (NP) number

---

[3]In other words, the average number of features active for a given token.

| Layer | % Variance Explained | L1 | L0 | % Alive | CE Diff | % CE Recovered |
|-------|---------------------|-----|-----|---------|---------|----------------|
| Resid 0 | 92% | 11 | 59 | 41% | 0.24 | 97% |
| Resid 1 | 85% | 13 | 54 | 38% | 0.45 | 95% |
| Resid 2 | 96% | 24 | 108 | 27% | 0.55 | 94% |
| Resid 3 | 96% | 23 | 68 | 22% | 0.58 | 95% |
| Resid 4 | 88% | 23 | 61 | 27% | 0.48 | 95% |
| Resid 5 | 90% | 35 | 72 | 45% | 0.55 | 92% |

Table 6: Residual (Resid) SAE evaluation by layer.

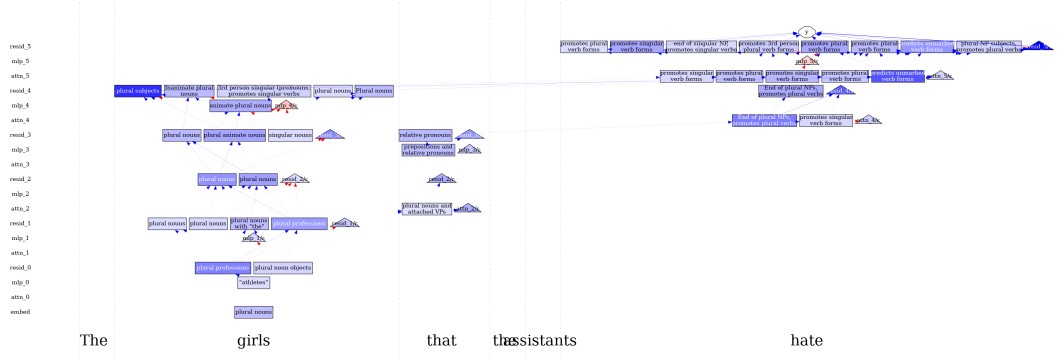

Figure 7: The feature circuit for agreement across a relative clause in Pythia-70M, computed using $T_N = 0.1$ and $T_E = 0.01$. The model detects the subject's number at the subject position. Other features detect relative pronouns (the start of the distractor clause). Finally, at the last token of the RC, the attention moves the subject information to the last position, where it assists in predicting the correct verb inflection.

tracking features, which are active at all positions for NPs of a given number (except on distractor phrases of opposite number). We present an example of an NP number tracker feature in Figure 15.

The circuits for agreement across a prepositional phrase (Figures 9 and 10) look remarkably similar to agreement across a relative clause; for both Pythia and Gemma, these two circuits share over 85% of their features, and many of the same features are used for detecting both prepositions and relative clauses.

For simple agreement (Figures 11 and 12), many of the same features that were implicated in noun number detection and verb number prediction in the previous circuits also appear here. The models detect the subject's number at the subject position in early layers. In later layers, these noun number detectors become inputs to verb number promoters, which activate on anything predictive of particular verb inflections.

The circuits for agreement within a relative clause (Figures 13 and 14) appear to have the same structure as that for simple agreement: subject number detectors in early layers, followed by verb number promoters in later layers.

## C.2 BIAS IN BIOS CIRCUIT

Here, we present the full annotated circuit discovered for the Bias in Bios classifier trained on Pythia-70M (described in §4 and App. E). The circuit was discovered using $T_N = 0.1$ and $T_E = 0.01$. We observe that the circuit (Figure 16) contains many nodes which simply detect the presence of gendered pronouns or gendered names. A few features attend to profession information, including one which activates on words related to nursing, and another which activates on passages relating to science and academia.

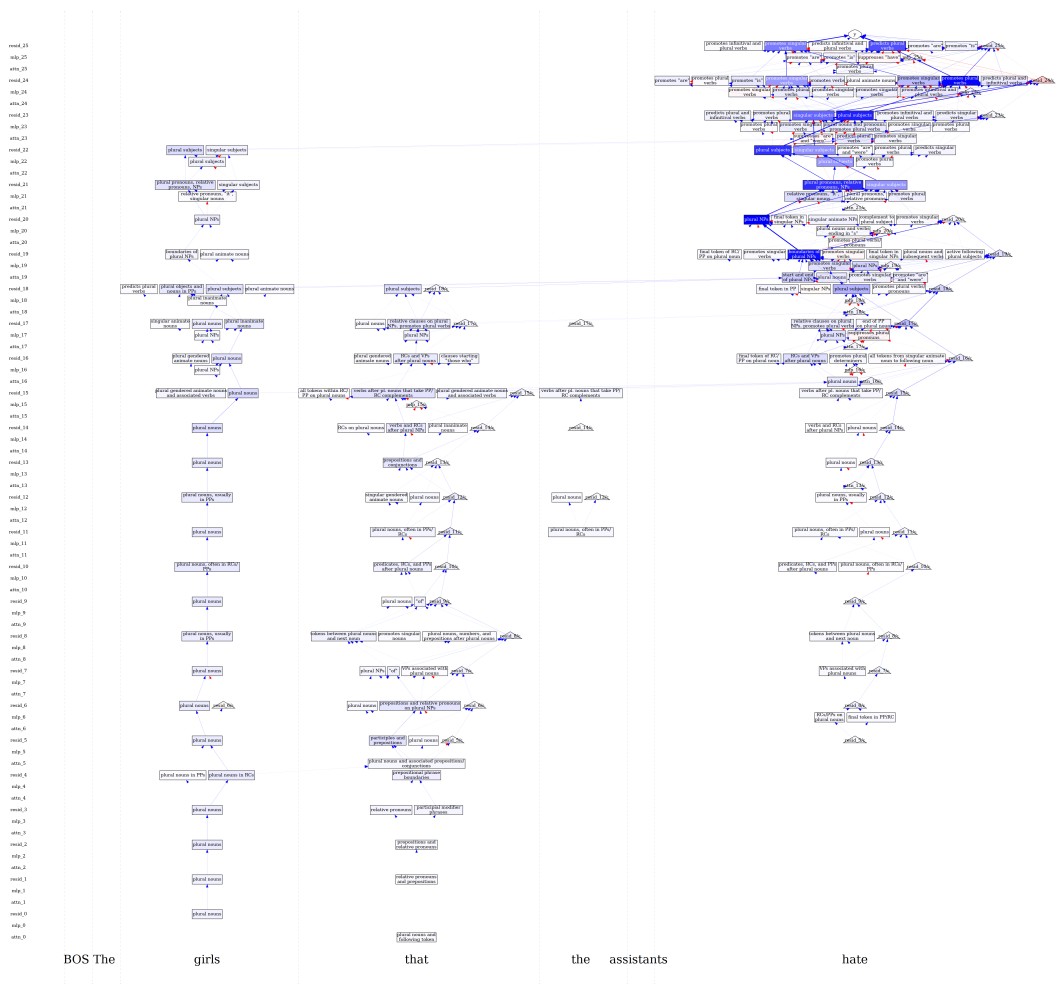

Figure 8: The feature circuit for agreement across a relative clause in Gemma-2-2B, computed using $T_N = 0.073$ and $T_E = 0.007$.

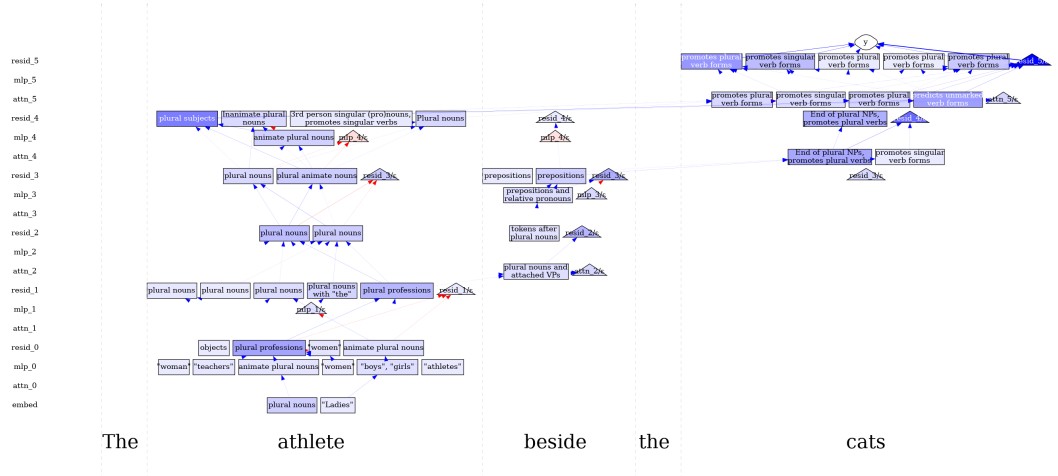

Figure 9: The feature circuit for agreement across a prepositional phrase in Pythia-70M, computed using $T_N = 0.1$ and $T_E = 0.01$. The model detects the subject's number at the subject position. Other features detect prepositional phrases (the start of the distractor clause). Finally, at the last token of the RC, the attention moves the subject information to the last position, where it assists in predicting the correct verb inflection.

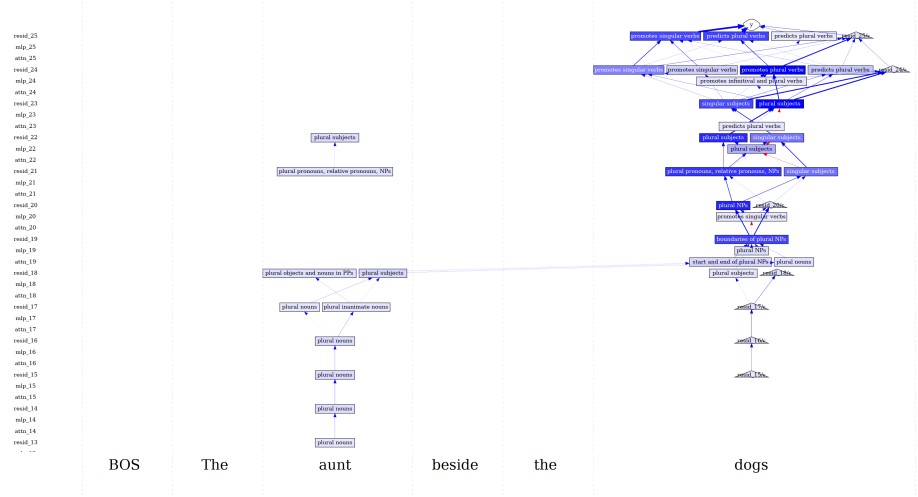

Figure 10: The feature circuit for agreement across a prepositional phrase in Gemma-2-2B, computed using $T_N = 0.5$ and $T_E = 0.05$. Note that we show the circuit beginning in layer 13, as our circuit discovery implicated only one node in earlier layers.

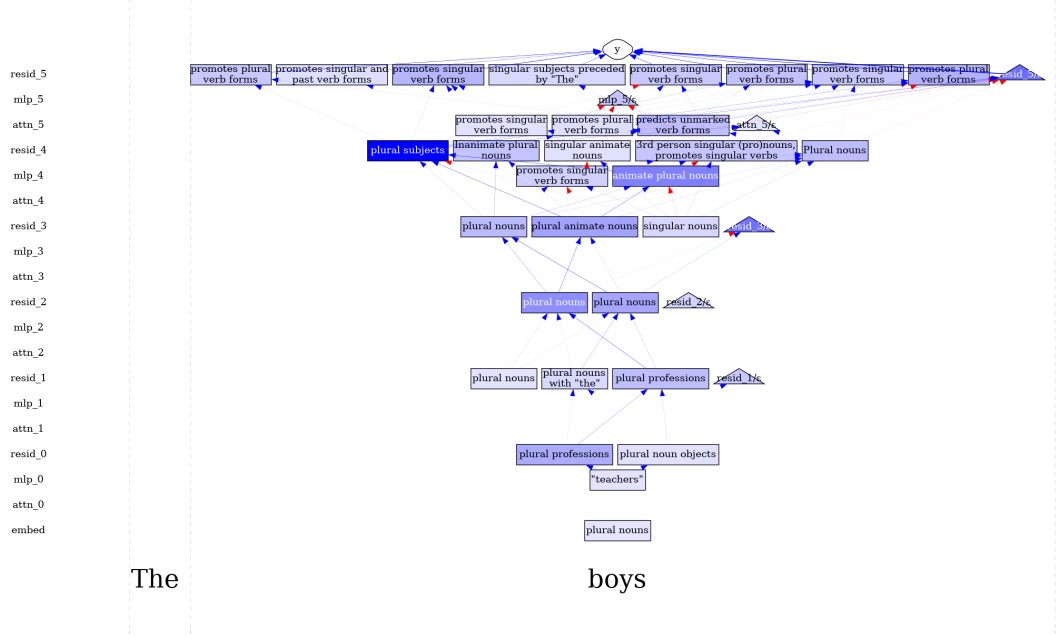

Figure 11: The feature circuit for simple agreement in Pythia-70M, computed using $T_N = 0.2$ and $T_E = 0.02$. The model detects the subject's number at the subject position in early layers. In later layers, these are inputs to features which activate on anything predictive of particular verb inflections.

## C.3 CLUSTER CIRCUITS

Here, we present full annotated circuits discovered for automatically discovered behaviors (described in App. G). First, we present the circuit for incrementing number sequences (Figure 17), discovered with $T_N = 0.4$ and $T_E = 0.04$. We note that this circuit includes many features which perform either succession (Gould et al., 2023) or induction (Olsson et al., 2022). The succession features in the layer 3 attention seem to be general; they increment many different numbers and letters (as in Figure 5). The induction features are sensitive only to specific tokens: for example, contexts of the form "$x3\ldots x3$", where "3" is a literal. These compose to form **specific successor** features in layer 5: the most strongly-activating layer 5 residual feature specifically increments "3" to "4" given induction-like lists, where each list item is preceded by the same string (e.g., "Chapter 1...Chapter 2...Chapter 3...Chapter").

The circuit for predicting infinitival objects (Figure 18, discovered with $T_N = 0.25$ and $T_E = 0.001$) contains two distinct mechanisms. First, the model detects the presence of specific verbs like "remember" or "require" which often take infinitival objects. Then, the model uses two separate mechanisms to predict infinitive objects. The first mechanism detects present-tense verbs, participles, or predicate adjectives which can be immediately followed by infinitival direct objects (e.g., "They were excited to..."). The second mechanism detects nominal direct objects that can directly precede infinitival object complements (e.g., "They asked us to..."). Finally, these two mechanisms both influence the output in layer 5 without fully intersecting.

## D SAMPLE FEATURES

### D.1 SPARSE FEATURES

Here, we present examples of sparse features with high indirect effects on the Bias in Bios task. Some of these features clearly activate on terms related to medicine or academia, which are related to the target profession classification task. Others simply detect the presence of "he" or female names.

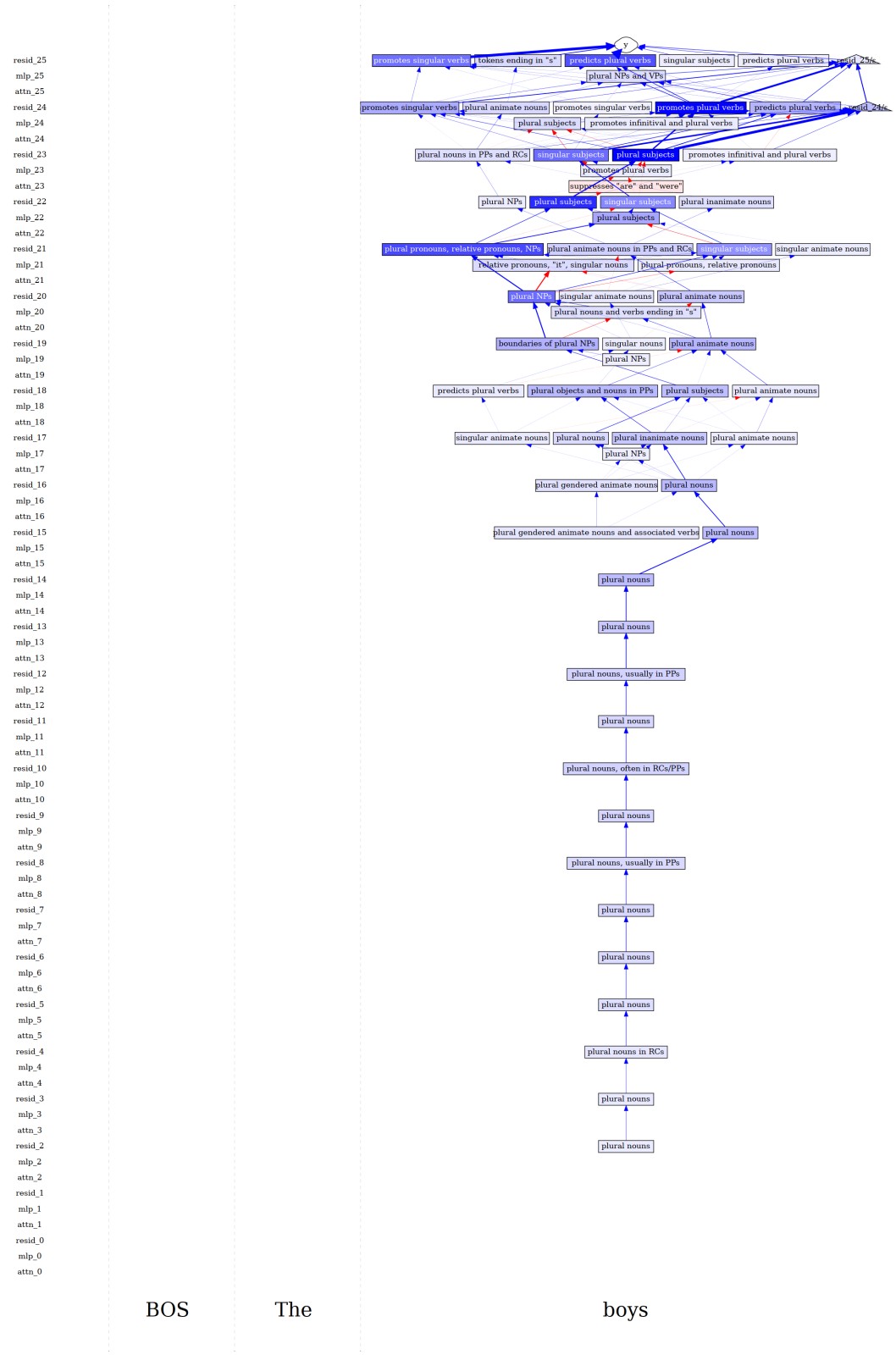

Figure 12: The feature circuit for simple agreement in Gemma-2-2B, computed using $T_N = 0.5$ and $T_E = 0.05$.

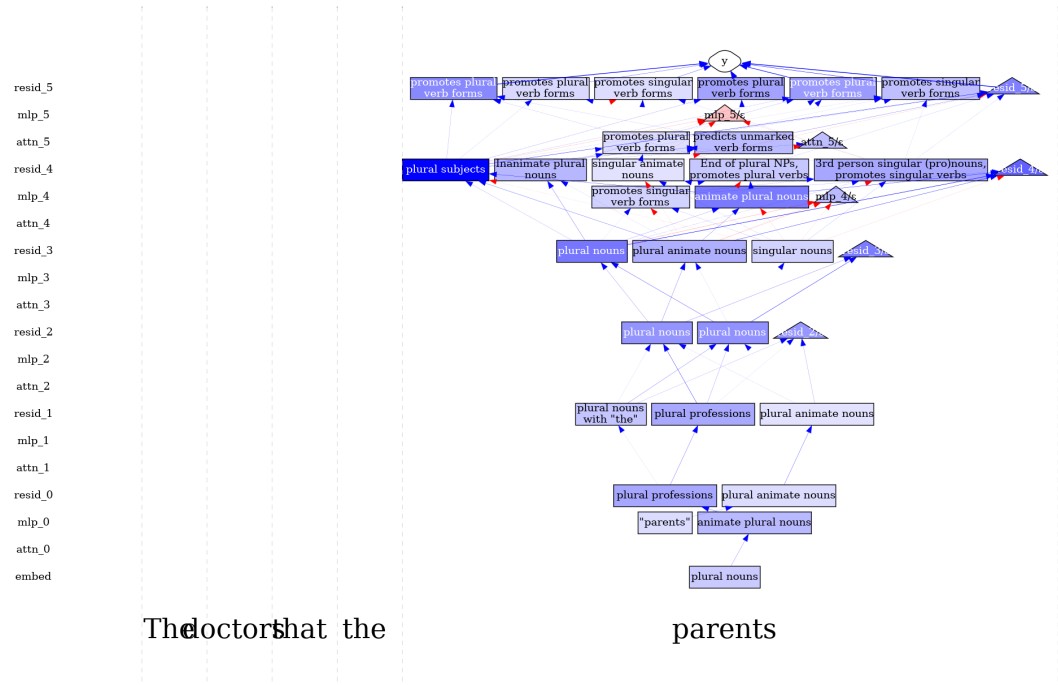

Figure 13: The feature circuit for agreement within a relative clause in Pythia-70M, computed with $T_N = 0.2$ and $T_E = 0.02$. The model detects the subject's number at the subject (within the RC)'s position in early layers. In later layers, these features are inputs to features which activate on anything predictive of particular verb inflections.

## D.2 NEURONS

For contrast, we also present examples of *dense* features—that is, neurons from MLPs, layer-end residuals, and the out-projection of the attention—with high indirect effects on the Bias in Bios task. We cannot directly interpret the activation patterns of these neurons, and so it is difficult to run the SHIFT with neurons baseline. We therefore instead compare to the neuron skyline, where we allow the skyline an unfair advantage by simply ablating neurons which have positive effects on gender-based probabilities given the balanced set.

## E IMPLEMENTATION DETAILS FOR CLASSIFIER EXPERIMENTS

### E.1 CLASSIFIER TRAINING

Here we describe how we train linear classification heads on Pythia-70M and Gemma-2-2B the Bias in Bios (BiB) task of §4.

Given a model $M$ and choice $\ell$ of layer, we mean-pool over (non-padding) tokens all layer $\ell$ residual stream activations from $M$; we then train a linear classification head via logistic regression, using the AdamW optimizer (Loshchilov & Hutter, 2017) and learning rate $0.01$ for one epoch on this dataset of activations. The activations and labels for this logistic regression are collected from the ambiguous set for the baseline classifier and from the balanced set for the oracle classifier.

To mimic a realistic application setting, we tune the choice $\ell$ of layer for the baseline probe's accuracy on (a test split of) the ambiguous set. For Pythia, this recommends using the penultimate layer $\ell = 4$. For Gemma, there is a wide range of equally performant layers. Thus—for this one choice only—we make use of the balanced set to compute how the baseline probe generalizes; we select the layer $\ell = 22$ for which the baseline probe generalizes *worst*. We make this choice to set up a testbed where there is the most space for improvement. We emphasize that we never tune

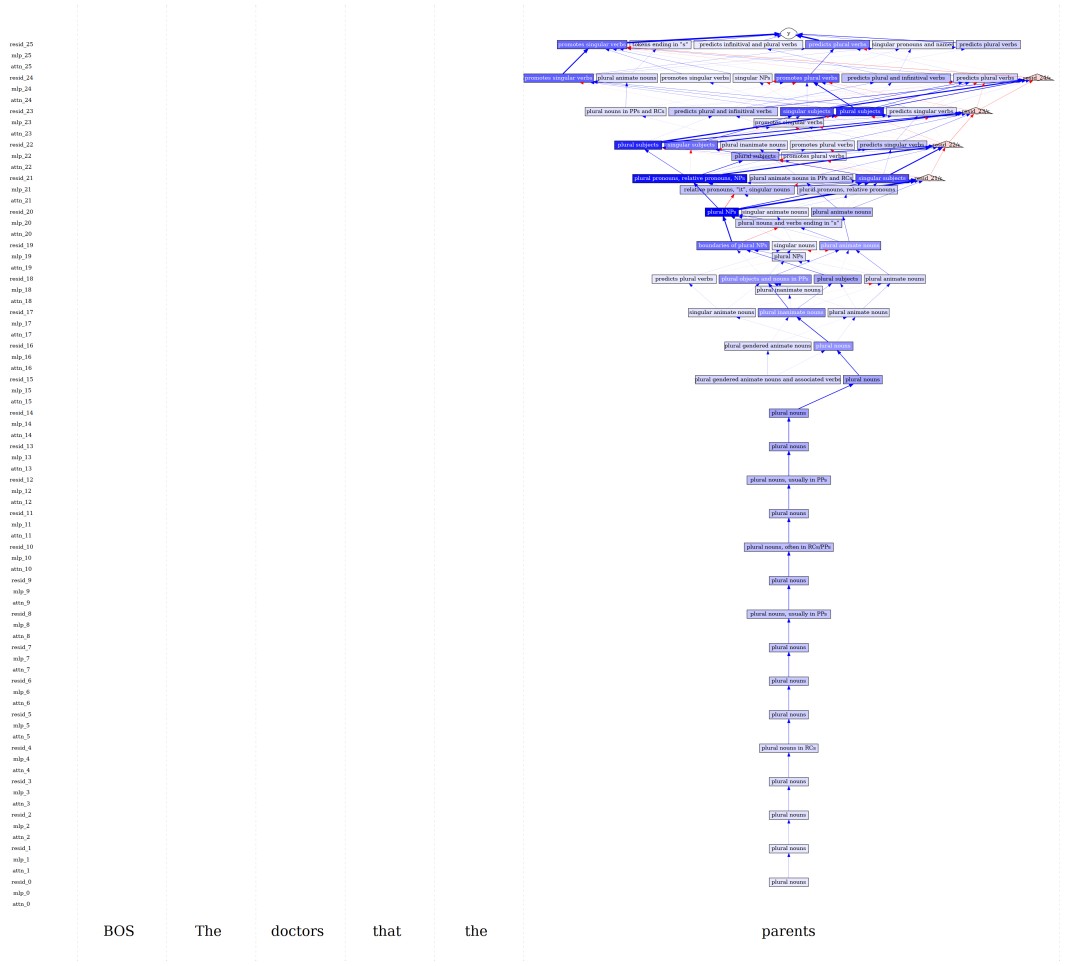

Figure 14: The feature circuit for agreement within a relative clause for Gemma-2-2B, computed with $T_N = 0.5$ and $T_E = 0.05$.

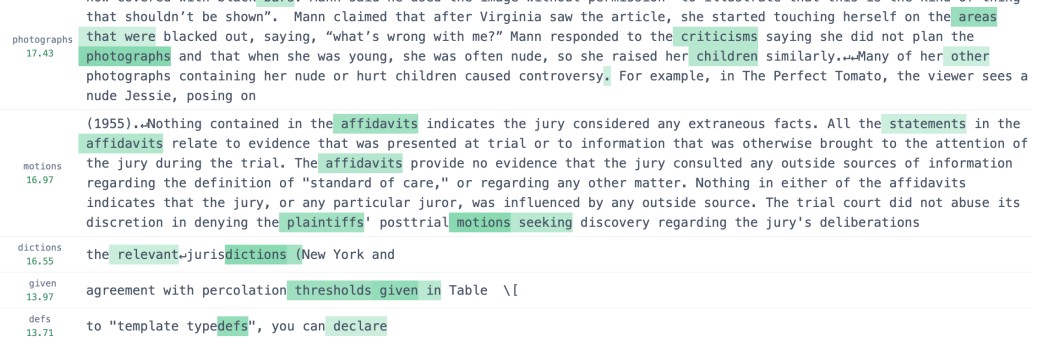

Figure 15: An example sparse feature for agreement across a relative clause in Gemma 2 (resid_12/13561). This feature activates on tokens in noun phrases where the noun head is plural, but not on singular distractor phrases within the plural NP. This feature carries the number of the subject across positions, so we term it an "NP number tracker".

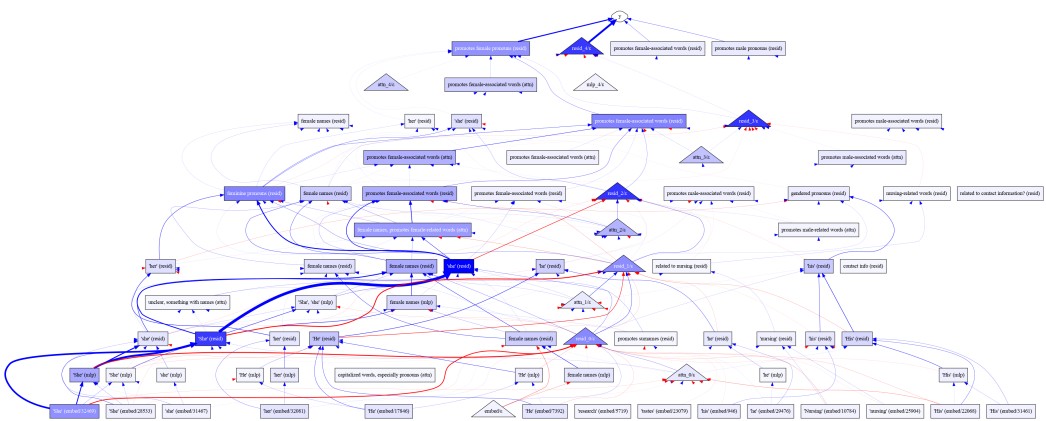

Figure 16: The full annotated feature circuit for the Bias in Bios classifier. Many nodes simply detect the presence of gendered pronouns or gendered names. A few features attend to profession information, including one which activates on words related to nursing, and another which activates on passages relating to science and academia.

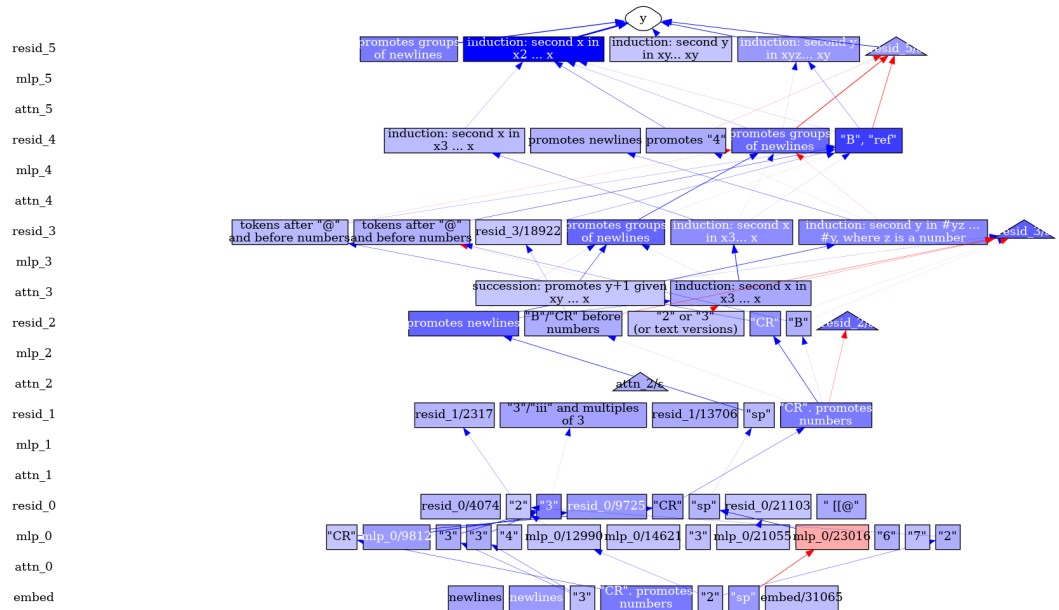

Figure 17: The full annotated feature circuit for incrementing number sequences. The model first detects the presence of specific number tokens, like "3". Later, it learns more robust semantic representations of those numbers, like "iii" and "Three". Then, the model uses a series of narrow and general succession and induction features to increment the next number.

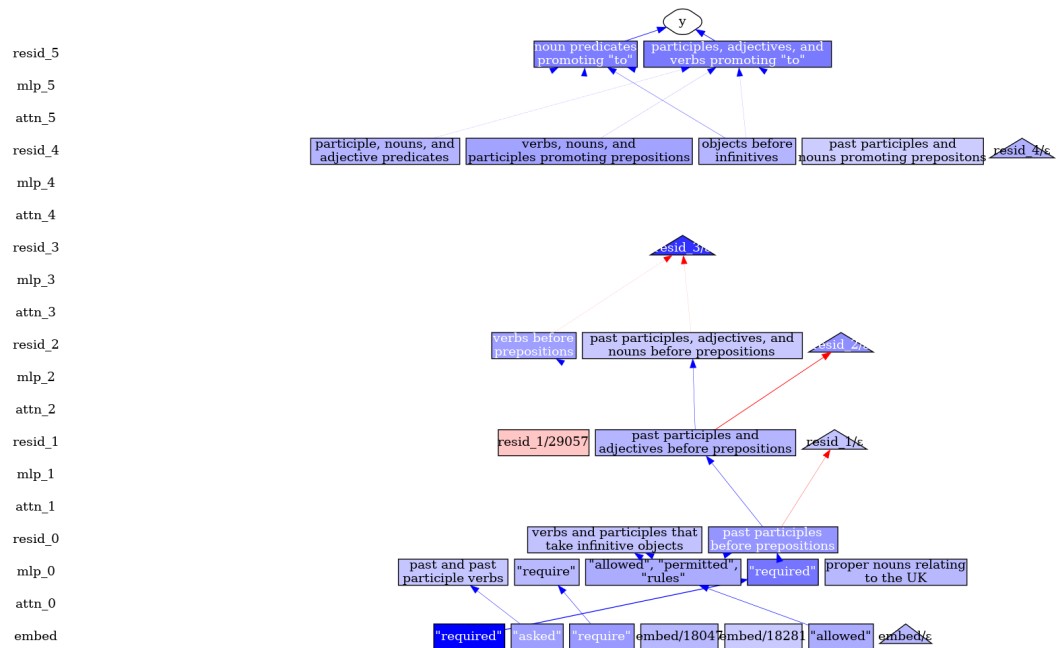

Figure 18: The full annotated feature circuit for predicting "to" as an infinitival object. The model first detects the presence of verbs that often take infinitival objects. Then, it uses one mechanism to detect present-tense verbs, participles, or predicate adjectives which take infinitival objects, and another mechanism to detect direct objects that can directly precede infinitival object complements. Finally, these two mechanisms both influence the output in layer 5 without fully intersecting.

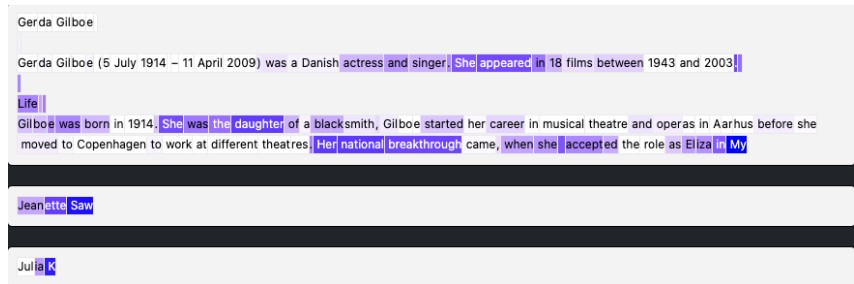

Figure 19: An example sparse feature from the Bias in Bios task (`attn_3/22029`). This feature detects female-related words in biographies of women. It also promotes words like "husband" and "née". This feature probably contributes to preferences for the spurious correlate of gender; we therefore ablate it.

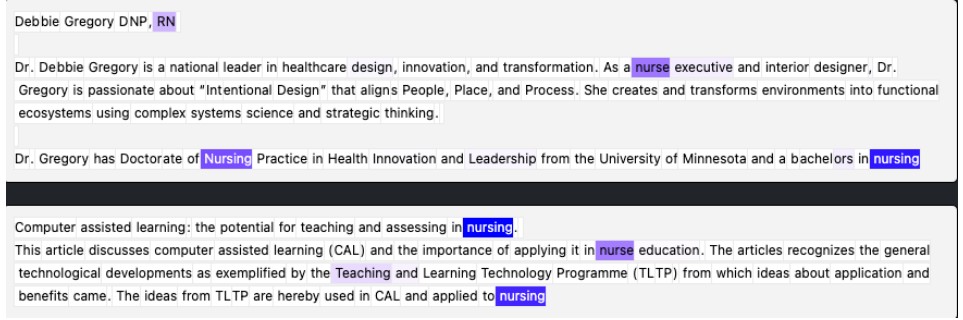

Figure 20: An example sparse feature from the Bias in Bios task (`resid_2/31098`). This feature activates on words related to nursing, including "RN" and "nurse". This probably relates to the target task of profession prediction. We therefore keep it.

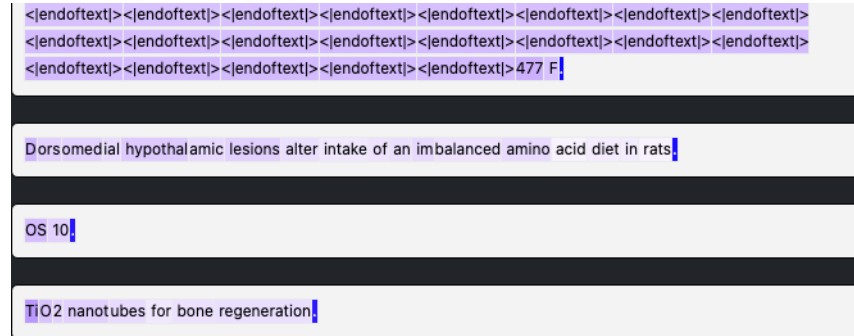

Figure 21: An example neuron from the Bias in Bios task. This appears to activate on beginnings and ends of sentences, but also more strongly on any token in a sentence that contains capital letters or numbers. We cannot deduce whether this would contribute more to gender or profession names.

Figure 22: An example neuron from the Bias in Bios task. This activates positively on tokens starting with capital letters, but negatively on many other tokens (whose unifying theme we cannot deduce).

hyperparameters for the performance **balanced set** of SHIFT, as using the **balanced set** is forbidden by the problem statement.

When retraining after performing SHIFT, we retrain *only* the linear classification head, not the full model.

### E.2 IMPLEMENTATION FOR CONCEPT BOTTLENECK PROBING

Our implementation for Concept Bottleneck Proving (CBP) is adapted from (Yan et al., 2023). It works as follows:

1. First, we collect a number of keywords related to the intended prediction task. We use $N = 20$ keywords: nurse, healthcare, hospital, patient, medical, clinic, triage, medication, emergency, surgery, professor, academia, research, university, tenure, faculty, dissertation, sabbatical, publication, and grant.
2. We obtain *concept vectors* $c_1, \ldots, c_N$ for each keyword by extracting Pythia-70M's penultimate layer representation over the final token of each keyword, and then subtracting off the mean concept vector. (Without this normalization, we found that concept vectors have very high pairwise cosine similarities.)
3. Given an input with representation $x$ (obtained via the mean-pooling procedure in App. E.1), we obtain a concept bottleneck representation $z \in \mathbb{R}^N$ by taking the cosine similarity with each $c_i$.
4. Finally, we train a linear probe with logistic regression on the concept bottleneck representations $z$, as in App. E.1.

We decided to normalize concept vectors but not input representations because it resulted in stronger performance. We also experimented with computing cosine similarities before mean pooling.

| Activation type | Interpretability |
|---|---|
| Dense (random) | 32.6 |
| Dense (agreement) | 30.2 |
| Dense (BiB) | 36.0 |
| Sparse (random) | 52.8 |
| Sparse (agreement) | 62.3 |
| Sparse (BiB) | 81.5 |

Table 7: Human interpretability ratings for dense (neuron) vs. sparse (autoencoder) features. We present mean interpretability scores across features on a 0–100 scale. We show scores for features that were either uniformly sampled (random), the top 30 by IÊ from the subject-verb agreement across RC task (agreement; §3.3), or the top 30 by IÊ for the Bias in Bios task (BiB; §4).

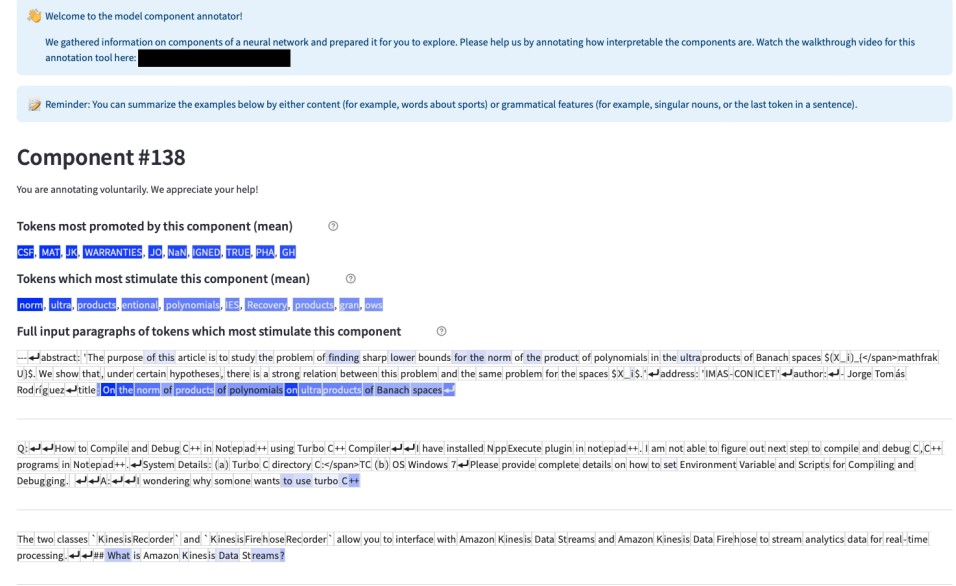

Figure 23: The human annotation interface used to obtain the interpretability ratings in Table 7. Here, we show the instructions, top-activating tokens, the token probabilities that were most affected when ablating the feature, and example contexts with feature activation values.

## F    HUMAN INTERPRETABILITY RATINGS FOR SPARSE FEATURES

Given our trained Pythia-70M sparse autoencoders, we asked human crowdworkers to rate the interpretability of random features, random neurons, features from our feature circuits, and neurons from our neuron circuits on a 0–100 scale (Table 7). Crowdworkers rate sparse features as significantly more interpretable than neurons, with features that participate in our circuits also being more interpretable than randomly sampled features.

See Figures 23 and 24 for examples of the human annotator interface. Humans were presented with the tokens on which the feature activated most strongly, followed by the tokens whose probabilities were most affected in Pythia-70M when the feature was ablated. This is followed by a series of example contexts in which the feature activated on some subset of tokens, where feature activations are shown in varying shades of blue (darker shades indicate higher activations). On the same page below the contexts, we ask annotators to write a textual description of the feature, and rate both its interpretability and its semantic complexity on 0–100 scales.

Crowdworkers were recruited from the ARENA Slack channel, whose members are machine learning researchers interested in AI alignment and safety. The selection of annotators certainly influ-

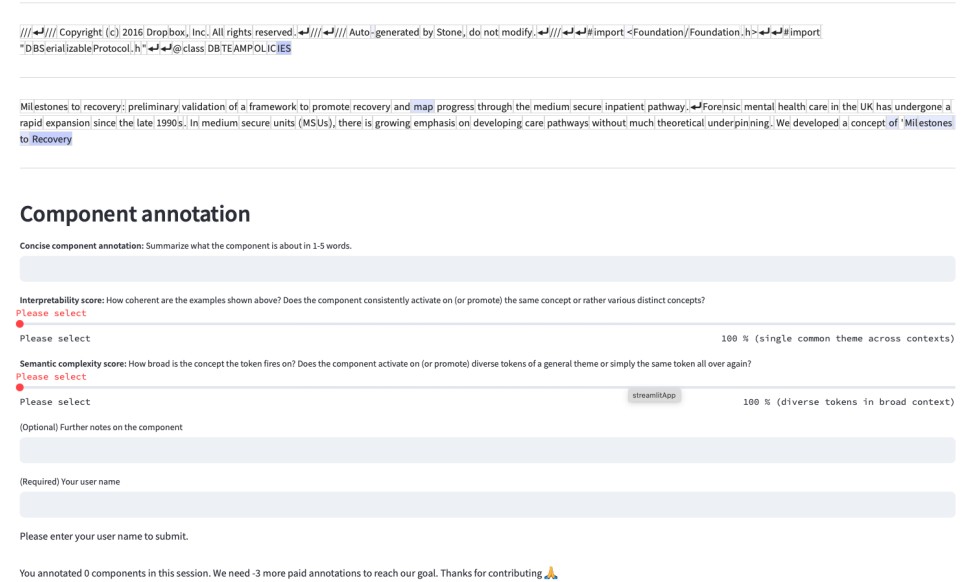

Figure 24: The human annotation interface used to obtain the interpretability ratings in Table 7. Here, we show the rating interface on the same page as the content in Fig. 23, below the example contexts. Humans were asked to write a textual description of each feature, assign a 0–100 interpretability rating, and assign a 0–100 semantic complexity rating to each feature.

enced our results; a truly random sample of human annotators would likely display higher variance when annotating features.

One common error pattern we notice is that annotators often label features according to semantic groupings (e.g., "text about politics," and do not pay attention to syntactic context (e.g., "plural nouns"). Future work could address this design bias by testing variants of the instructions.

Results of human evaluations for the Gemma Scope SAEs are described in Lieberum et al. (2024).

## G  DISCOVERING LM BEHAVIORS WITH CLUSTERING

In this section, we describe our unsupervised method for discovering language model behaviors. More specifically, following Michaud et al. (2023), we cluster contexts from The Pile according to the Pythia-70M's internal state during inference. In this section, we describe our clustering pipeline and methods.

### G.1  FILTERING TOKENS

We must first locate (context, answer) pairs for which an LM correctly predicts the answer token from the context. We select The Pile (Gao et al. (2020)) as a general text corpus and filter to pairs on which Pythia-70M confidently and correctly predicts the answer token, with cross-entropy lower than 0.1 or 0.3 nats, depending on the experiment. The model consistently achieves low loss on tokens which involve "induction" (Olsson et al., 2022)—i.e., tokens which are part of a subsequence which occurred earlier in the context. We exclude induction samples by filtering out samples in which the bigram (final context token, answer token) occured earlier in the context.

### G.2  CACHING MODEL-INTERNAL INFORMATION

We find behaviors by clustering samples according to information about the LM's internals when run on that sample. We find clusters of samples where the model employs similar mechanisms for next-token prediction. We experiment with various inputs to the clustering algorithm:

- **Dense Activations**: We take activations (residual stream vectors, attention block outputs, or MLP post-activations) from a given context and concatenate them. To obtain a vector whose length is independent of the context length, we can either use the activations at the last $N$ context positions before the answer token, or aggregate (sum) across the sequence dimension. We experiment with both variants.

- **Sparse Activations**: Rather than dense model activations, we can use the activations of SAE features. We concatenate and aggregate these in the same manner as for dense activations.

- **Dense Component Indirect Effects**: We approximate the indirect effect of all features on the correct prediction using 2 without a contrastive pair—namely, by setting $\mathbf{a}_{\text{patch}} = 0$. The negative log-probability of the answer token $m = -\log p(answer)$ serves as our metric for the correct prediction of the next token. The computatiom of linear effects requires saving both (1) activations and (2) gradients w.r.t $m$ at the final $N$ positions for each context in the dataset. We optionally aggregate by summing over all positions.

- **Sparse Indirect Effects**: Similarly, we can compute the linear effects of *sparse* activations on the correct prediction.

- **Gradient w.r.t. model parameters**: As in Michaud et al. (2023), we also experiment with using gradients of the loss w.r.t. model parameters, but with some modifications. We describe this method in more detail in §G.3 below.

### G.3 HYPERPARAMETERS AND IMPLEMENTATION DETAILS

We apply either spectral clustering or $k$-means clustering. For spectral clustering, given either activations or effects $x_i$ for sample $i$, we compute a matrix of pairwise cosine similarities $C_{ij} = x_i \cdot x_j/(||x_i||||x_j||)$ between all pairs of samples. Before performing spectral clustering, we normalize all elements of $C$ to be in $[0, 1]$ by converting the cosine similarities to angular similarities: $\hat{C}_{ij} = 1 - \arccos(C_{ij})/\pi$.

We use the `scikit-learn` (Pedregosa et al., 2011) spectral clustering implementation with $k$-means. For all inputs except gradients w.r.t. model parameters, we used spectral clustering across 8192 samples. We chose $k$ (the number of total clusters) to maximize the number of clusters implicated in more than one input context.

We also experimented with using gradients w.r.t. model parameters as inputs, as in Michaud et al. (2023). Here, we scale up our approach to 100,000 samples. It is intractable to perform spectral clustering given 100,000 samples, so we instead use $k$-means clustering. Rather than clustering the gradients themselves (which are high-dimensional), we cluster sparse random projections of the gradients down to 30,000 dimensions. When projecting, we use a matrix with entries $\{-1, 0, 1\}$. When sampling the entries of this matrix, sample a nonzero value with probability $32/30000$, and if nonzero, sample $-1$ or $1$ with equal probability. For a sparse projection matrix with dimensions $\mathbb{R}^{n \times 30000}$, there will on average be $32 \cdot n$ nonzero entries, where $n$ is the number of parameters in the model.[4]

## H QUALITY OF LINEAR APPROXIMATIONS OF INDIRECT EFFECTS

Figure 25 shows the quality of our linear approximations for indirect effects. Prior work (Nanda, 2022; Kramár et al., 2024) investigated attribution patching accuracy for IEs of coarse-grained model components (queries, keys, and values for attention heads, residual stream vectors, and MLP outputs) and MLP neurons. Working with SAE features and errors, our results echo previous findings: attribution patching is generally quite good, but sometimes underestimates the true IEs. Notable exceptions are the layer 0 MLP and the residual stream in early layers. We also find that our integrated gradients-based approximation significantly improves approximation quality.

---

[4]We only consider gradients w.r.t. non-embedding and non-layernorm parameters.

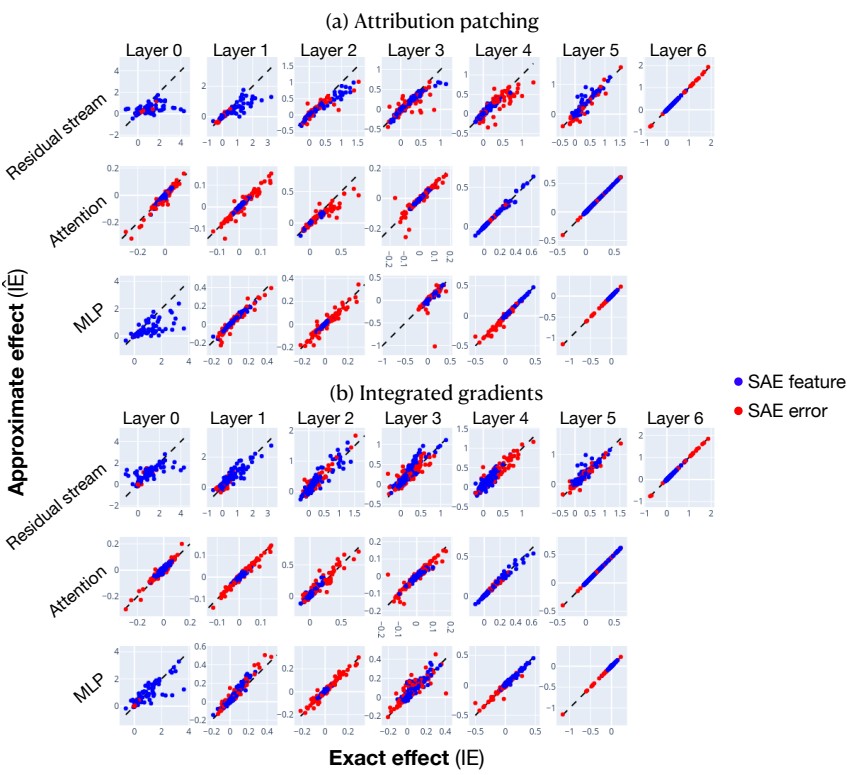

Figure 25: Approximate IEs ($y$-axis) and exact IEs ($x$-axis) using attribution patching (a; top) or integrated gradients (b; bottom). Each point corresponds to an SAE feature or SAE error at one token position of one input. Data were collected from 30 inputs from our across RC dataset.

