# OpenReview forum: "Sparse Feature Circuits: Discovering and Editing Interpretable Causal Graphs in Language Models"
_ICLR.cc/2025/Conference — ICLR 2025 Oral_

### Official Review · Reviewer_teh3 · 2024-11-02

**Soundness:** 3
**Presentation:** 4
**Contribution:** 4
**Rating:** 8
**Confidence:** 3

**Summary:**

This paper introduces a method that automatically discovers "sparse feature circuit", defined as subgraphs of a large language model that satisfy two desiderata: (1) responsible for implementing a model behavior, and (2) human-interpretable. As a by-product, the authors propose SHIFT, a method that removes non-causal features (e.g. gender bias) from a classifier. Finally, the authors scale their method to thousands of automatically discovered behaviors from SAEs, hinting at the possibility of automated interpretability.

**Strengths:**

- This paper is very well written, and provides enough background on SAEs to motivate their approach. Each component of the method is also presented very clearly.
- This contribution is very timely. Most prior works on mechanistic interpretability suffers from two pitfalls: (1) Inability to understand / interpret dense MLP layers [1], and (2) what's next after interpretability. This work provides satisfactory answers to both, by (1) using more recent approaches of SAEs to disentangle polysemanticity and (2) using interpretable features to debias a classifier / remove non-causal relationships from it.
- Owning to the scalability of SAEs, this method can be scaled to automatically discover thousands of behaviors to billion scale parameters. The authors also supplement this ability with an interactive website, which offers many interesting insights and improves accessibility of this research.

[1] How does GPT-2 compute greater-than?: Interpreting mathematical abilities in a pre-trained language model, https://arxiv.org/abs/2305.00586

**Weaknesses:**

- While this approach is scalable with respect to model size and a broad range of behaviors, using this approach would still require contrastive sample pairs that is hard to get and / or define for tasks such as long-form generation and algorithmic reasoning. Nevertheless, I believe this work represents a step in the right direction, but it'd be great for the authors to discuss this type of dataset limitations and future directions to automatically construct contrastive pairs for more challenging tasks.
- As the authors have mentioned, most of these results are qualitative, and it would be great to establish a more quantitative benchmark to supplement the other interesting behaviors that the authors were able to mine from their unsupervised approach.

**Questions:**

See weaknesses

---

> ### Author Response · Authors · 2024-11-21
> **Response**
>
> Thank you for the positive feedback! We are glad that you found our method timely, scalable, and clearly presented. We address your concerns and questions below.
>
> > …[U]sing this approach would still require contrastive sample pairs that is hard to get and / or define for tasks such as long-form generation and algorithmic reasoning… [I]t'd be great for the authors to discuss this type of dataset limitations and future directions to automatically construct contrastive pairs for more challenging tasks.
>
> Note that we do not necessarily require contrastive sample pairs! See our experiments on unsupervised circuit discovery, where no contrastive pairs were used, and moreover no human needed to make decisions about task definition and data selection.
>
> Regarding long-form generation, you are correct that this paper focuses on circuits which explain the prediction of particular tokens. However, the core ideas should generalize: instead of using the log-probability of a single token as the metric to explain, one could instead use the log-probability of a multi-token completion. This substitution would be a drop-in replacement—nothing else about the method would need to change.
>
> > As the authors have mentioned, most of these results are qualitative, and it would be great to establish a more quantitative benchmark to supplement the other interesting behaviors that the authors were able to mine from their unsupervised approach.
>
> We agree! We have work in progress on this; stay tuned!

---

> > ### Comment · Reviewer_teh3 · 2024-11-22
> >
> > Thank you for your clarifications. Since my review is already positive and the weaknesses didn't result in any reduction in rating during my review, I am keeping the score as is. Keep up the good work :-)

---

### Official Review · Reviewer_xPLM · 2024-11-02

**Soundness:** 3
**Presentation:** 3
**Contribution:** 4
**Rating:** 8
**Confidence:** 3

**Summary:**

This paper introduces a method for discovering interpretable sparse feature circuits in LLMs using sparse autoencoders (SAEs). While previous mechanistic interpretability work focused on polysemantic neurons, MLP attention heads, this paper shows how to decompose model behaviors into monosemantic, human-interpretable features. The authors validate their approach through three key contributions: (1) a circuit discovery method combining SAEs with linear approximations of indirect effects, (2) a debiasing application called SHIFT that ablates task-irrelevant features, and (3) demonstration of scalable, automated circuit discovery for large sets of model behaviors.

**Strengths:**

Strengths:

Successfully addresses a core challenge in MI: finding and scaling interpretable units of analysis.


- Empirical Results:

Comprehensive subject-verb agreement circuits reveal clear compositional structure (e.g., distinct pathways for handling relative clauses vs. prepositional phrases)
Human evaluation shows significantly better feature interpretability (81.5 for BiB circuit features vs 36.0 for neurons)
SHIFT achieves 93.1% profession accuracy while reducing gender bias to near chance (52.0%)

- Validation & Reproducibility:

Thorough ablation studies on all key parameters (SAE width, layer selection, manipulation coefficients). Clear comparison against neuron-based approaches showing consistent advantages.

**Weaknesses:**

Weaknesses:

- No code and data released yet.

- Theoretical Analysis:

In terms of theoretical weaknesses, some formal characterization of when SAE features capture meaningful concepts vs memorized patterns would strength the paper.


Most of these are noted in the paper but it's worth mentioning it here, SAE training requires significant compute (2 billion tokens) with potential instability (dead features). The current implementation limited to decoder-only models (Pythia-70M and Gemma-2-2B).

- Evaluation Scope:

Subject-verb agreement circuits focus on relatively simple grammatical phenomena and this might not be an issue with scale but no mention of anything more complex in the paper.

**Questions:**

- The paper shows strong results with linear approximations of indirect effects. Have you checked when/why these approximations break down? This seems especially relevant for early layers where you note integrated gradients improves accuracy.

- For the automated circuit discovery, how robust are the discovered clusters to different choices of projection dimension and clustering hyperparameters?

---

> ### Author Response · Authors · 2024-11-21
> **Response to Weaknesses**
>
> Thank you for the positive feedback! We are glad you found our work to address core challenges in interpretability, and that you found our experiments to be effective and clear. We address your concerns and questions below.
>
> Weaknesses
> ===
> > No code and data released yet.
>
> Please note that our code and data are submitted as supplementary material. These will be publicly released after the review period.
>
> > …some formal characterization of when SAE features capture meaningful concepts vs memorized patterns would strength the paper.
>
> To clarify: By "memorized patterns", are you referring to (i) features that are used by the model, but which are uninteresting because they capture memorized token associations, or (ii) cases where the SAE itself has memorized training data in a way that does not correspond to something learned by the language model? We note that features of type (i) may in fact play an important role in a model’s computation (e.g. the association between “Michael Jordan” and “basketball” is inherently a memorized one), so should rightfully appear when explaining model behaviors. Thus, we will assume that you are concerned about (ii).
>
> For this, causal verification provides a way forward. If we annotate a feature as being related to concept C, then we can confirm that modifying the network’s computation by intervening on the feature affects the model’s output in a predictable way related to C. While we don’t do a thorough analysis of this for all features in our SAEs, our SHIFT experiment provides some validation: we see that intervening on networks by removing their gender-related features causes their activations to encoder less gender-related information.
>
> > SAE training requires significant compute (2 billion tokens) with potential instability (dead features). The current implementation limited to decoder-only models (Pythia-70M and Gemma-2-2B).
>
> We agree. We will note that, based on SAE scaling laws [1] that were made available after this paper’s preparation, our Pythia-70M SAEs were actually overtrained by a factor of 10. So it would have actually been possible to produce SAEs of a similar quality with substantially fewer training tokens. There’s also been significant progress on reducing the fraction of dead features [2, 3].
>
> > Subject-verb agreement circuits focus on relatively simple grammatical phenomena and this might not be an issue with scale but no mention of anything more complex in the paper.
>
> We agree that it would be interesting to apply sparse feature circuits (SFCs) to more complex tasks. If this concern is primarily on the diversity of tasks, we believe that the unsupervised interpretability pipeline partially addresses this point: it reveals that SFCs allow one to tractably interpret very diverse behaviors in LMs. (Please feel free to explore these diverse behaviors at our demo at `feature-circuits.xyz`.) The Bias in Bios experiment also reveals that sparse feature circuits are effective not just in revealing interpretable mechanisms underlying syntactic agreement tasks, but also those underlying a semantic classification task.

---

> ### Author Response · Authors · 2024-11-21
> **Response to Questions**
>
> Questions
> ===
> > The paper shows strong results with linear approximations of indirect effects. Have you checked when/why these approximations break down? This seems especially relevant for early layers where you note integrated gradients improves accuracy.
>
> In general, the linear approximations were worse in earlier layers, and especially for the layer 0 MLP; see figure 25 in App. H. Intuitively, we expect that linear approximations are worse for earlier layers because there are more nonlinearities between early layers and the model’s output. (As an extreme case in the opposite direction, the model’s output is a linear function of its final layer, so linear approximations are perfect for the final layer.)
>
> While not surprising, we’ll note explicitly that approximation error doesn’t seem to be any worse due to using SAE features. Our results on the efficacy of linear approximations (and when they fail) are generally in line with prior work, and Figure 25 does not show different trends for SAE features vs. error terms.
>
> > For the automated circuit discovery, how robust are the discovered clusters to different choices of projection dimension and clustering hyperparameters?
>
> As with our other results on unsupervised circuit discovery, we lack the metrics to quantitatively analyze the thousands of circuits produced by our unsupervised pipeline. However, we did try six different approaches to clustering, and we’re happy to give some qualitative takeaways. If you’d like, you can browse the clusters and corresponding feature circuits for all of these variants at feature-circuits.xyz.
>
> One axis of variation was clustering based on activations vs. based on linear indirect effects (i.e. activations * gradients). It generally seemed like the former method produced clusters based on similar inputs, whereas the latter produced some clusters based on similar outputs (e.g. clusters typified by the model’s output always being a particular token).
>
> Another axis of variation was how to aggregate activations/linear effects across various tokens. The options we explored were: only take the activation/indirect effect over the final token; over the final 5 tokens; or sum over all tokens. The difference between final token vs. final 5 tokens was what you might expect: the final-token clusters seemed to more strongly depend on the last token of the input. Surprisingly, for linear effects, we didn’t observe much of a difference between just using the final token or summing over all tokens; it seems that (consistently with our circuits), when doing causal attribution for the next-token-prediction, activations in the final token position consistently have the largest causal effects.
>
> We also replicated clustering via parameter gradients as in [4]. These clusters looked qualitatively similar to our last-token activation or linear effect clusters.
>
> We didn’t experiment with different choices of projection dimension. We chose the projection dimension to approximately preserve the unprojected cosine similarities, which seemed like a principled choice.
>
> References
> ===
> [1] Lindsey, et al. (2024). Scaling Laws for Dictionary Learning. https://transformer-circuits.pub/2024/april-update/index.html#scaling-laws
>
> [2] Gao et al. (2024). Scaling and Evaluating Sparse Autoencoders. https://arxiv.org/abs/2406.04093
>
> [3] Conerly et al. (2024). Update on how we train SAEs. https://transformer-circuits.pub/2024/april-update/index.html#training-saes
>
> [4] Michaud et al. (2023). The Quantization Model of Neural Scaling. https://openreview.net/forum?id=3tbTw2ga8K

---

> > ### Comment · Reviewer_xPLM · 2024-11-22
> > **Response to Authors (Thank you!)**
> >
> > Thank you for answering my questions and comments. Almost all of my comments and concerns are answered. I want to thank the authors for this impressive work and I am happy to maintain my score I would encourage the other reviewers to adjust their scores to reflect the quality and importance of this work.

---

### Official Review · Reviewer_gvvK · 2024-11-04

**Soundness:** 4
**Presentation:** 3
**Contribution:** 3
**Rating:** 8
**Confidence:** 5

**Summary:**

This paper presents Sparse Feature Circuits, a method to discover interpretable causal graphs within language models (LMs) based on sparse, human-interpretable features. Leveraging sparse autoencoders (SAEs), the authors identify feature circuits that reveal causally implicated components in model behaviors, contrasting with prior work focused on more complex components like attention heads and neurons. The technique allows for targeted interventions to refine model predictions by eliminating task-irrelevant features, as demonstrated in a new method called SHIFT. SHIFT enables selective feature removal, improving classifier generalization without requiring disambiguating labels. The paper’s contributions include a scalable pipeline for automatic discovery of interpretable circuits, and applications in improving model behavior by editing these circuits.

**Strengths:**

- By focusing on sparse feature circuits instead of larger, polysemantic components, this approach introduces a new level of granularity in LM interpretability. The use of SAEs is particularly innovative, as it allows for fine-grained, interpretable features that are scalable and can be applied across thousands of discovered behaviors, improving previous interpretability efforts that were limited by reliance on human-defined hypotheses.

- The SHIFT technique, designed to improve model generalization by removing unintended signals, offers a practical, actionable method for handling spurious features. The fact that SHIFT can operate without disambiguating labels and apply human judgment to identify spurious features highlights its practical utility for real-world model refinement, especially in settings where unintended biases are a concern.

- The unsupervised interpretability pipeline for discovering feature circuits across various model behaviors (including grammatical tasks like subject-verb agreement) is impressive. It enables scalable, automatic generation of feature circuits, addressing the labor-intensive nature of previous interpretability work. This unsupervised aspect expands the approach’s applicability across domains and data types, making it a versatile contribution.

- The paper presents quantitative evaluations on interpretability metrics such as faithfulness and completeness for both feature and neuron circuits, providing objective assessments of the interpretability advantage of sparse features.

**Weaknesses:**

- The method relies on human judgment to determine which features are spurious, which may lead to inconsistencies across applications or datasets. The framework lacks a clear evaluation on the repeatability and consistency of human-selected spurious features in SHIFT, especially when applied by different users or in different contexts. Including a quantitative study or qualitative analysis on the consistency of feature selection across different users or scenarios would strengthen the reliability of SHIFT’s results and enhance confidence in its practical application. For instance, assessing inter-annotator agreement or providing a standardized checklist could help mitigate inconsistencies.

- Although SHIFT leverages human judgment for feature interpretation, this reliance could hinder scalability in complex or high-dimensional models, as manual inspection becomes increasingly challenging. The paper does not provide clear guidelines on how to efficiently scale this process for larger models or more diverse tasks.

- The method’s success is heavily tied to the availability and quality of sparse autoencoders (SAEs). If the SAEs are suboptimal, the interpretability and causal circuit discovery might suffer, particularly if they fail to capture meaningful latent structures. This dependency on pretrained SAE quality raises questions about the method’s consistency and applicability across different model architectures or under domain shifts.

- The faithfulness and completeness metrics for feature circuits are assessed primarily on grammatical tasks like subject-verb agreement, which may not generalize to more complex language tasks that involve nuanced semantic understanding or context-sensitive behaviors. This limited evaluation could affect confidence in the generalizability of the results. Expanding evaluations to include complex, semantically challenging tasks (e.g., multi-turn dialogue or reasoning tasks) would provide stronger evidence of the method’s robustness.

**Questions:**

1. How do the authors ensure consistency in identifying spurious features across different annotators or applications? Are there any plans to standardize the interpretation criteria?

2. Could the authors clarify how the performance of SAEs affects the interpretability and effectiveness of feature circuits? Have they tested their approach on varying SAE architectures to assess robustness?

3. For large-scale models with complex circuits, what optimizations or techniques do the authors recommend to maintain interpretability while managing the increased number of features?

---

> ### Author Response · Authors · 2024-11-21
> **Response Pt. 1**
>
> We are grateful that you found our technique to be innovative, useful, and scalable! We address your concerns and questions below.
>
> Weaknesses
> ===
> > The framework lacks a clear evaluation on the repeatability and consistency of human-selected spurious features in SHIFT, especially when applied by different users or in different contexts.
>
> This is a valid point. We have added a statement to the limitations about this.
>
> > Although SHIFT leverages human judgment for feature interpretation, this reliance could hinder scalability in complex or high-dimensional models, as manual inspection becomes increasingly challenging. The paper does not provide clear guidelines on how to efficiently scale this process for larger models or more diverse tasks.
>
> There has been some work on automatically labeling sparse features using LLMs [1,2]. While not perfect, we believe that this line of work has significant potential to automate away much of the human labor involved in performing SHIFT. In particular, the annotation task involved in our SHIFT experiments was actually relatively simple, in the sense that it was almost always very easy to tell whether a feature was related to gender; see the example features in Figures 19 and 20. Once a human has identified gender as an unintended signal of interest from inspecting a few features, we think it’s likely that a well-prompted LLM would be effective at systematically searching through the full collection of features for gender-related ones.
>
> > The method’s success is heavily tied to the availability and quality of sparse autoencoders (SAEs). If the SAEs are suboptimal, the interpretability and causal circuit discovery might suffer, particularly if they fail to capture meaningful latent structures. This dependency on pretrained SAE quality raises questions about the method’s consistency and applicability across different model architectures or under domain shifts.
>
> This is a valid point. We rely on high-quality SAEs to discover useful and interpretable sparse feature circuits. That said, the SAE training techniques available during this paper’s preparation were already good enough to enable applications like the ones we demonstrate, and there has been rapid progress since then [3,4].
>
> As a secondary point, because our technique computes circuits containing SAE error nodes and quantifies the causal importance of these nodes, it is well-suited to be adapted into a metric for guiding future progress in SAEs. We leave exploring this direction to future work.
>
> > This limited evaluation could affect confidence in the generalizability of the results. Expanding evaluations to include complex, semantically challenging tasks (e.g., multi-turn dialogue or reasoning tasks) would provide stronger evidence of the method’s robustness.
>
> We agree that it would be interesting to apply sparse feature circuits (SFCs) to highly capable models on difficult tasks. If this concern is primarily on the diversity of tasks, we believe that the unsupervised interpretability pipeline partially addresses this point: it reveals that SFCs allow one to tractably interpret very diverse behaviors in LMs. (Please feel free to explore these diverse behaviors at our demo at `feature-circuits.xyz`.) The Bias in Bios experiment also reveals that sparse feature circuits are effective not just in revealing interpretable mechanisms underlying syntactic agreement tasks, but also those underlying a more semantics-based classification task.
>
> If this concern is instead on the difficulty of the tasks, we agree that it would be interesting to analyze the mechanisms underlying performance on datasets used to benchmark state-of-the-art models. For example, a mechanistic analysis of how Llama 3.1 (405B) accomplishes ARC and MMLU could likely be its own paper!

---

> ### Author Response · Authors · 2024-11-21
> **Response Pt. 2**
>
> Questions
> ===
> > How do the authors ensure consistency in identifying spurious features across different annotators or applications? Are there any plans to standardize the interpretation criteria?
>
> We had two authors independently annotate each sparse feature. We then compared annotations and resolved differences via discussion. In practice, it is likely true that different annotators will arrive at distinct annotations. However, as discussed above, we believe that if this were reduced to the binary classification task of simply assessing whether a feature is relevant or spurious to the task, then there would be significantly more inter-annotator agreement.
>
> > Could the authors clarify how the performance of SAEs affects the interpretability and effectiveness of feature circuits? Have they tested their approach on varying SAE architectures to assess robustness?
>
> The interpretability, faithfulness, and completeness of discovered feature circuits crucially rely on SAE quality, as discussed above. Please note that the Pythia-70M and Gemma-2-2B SAEs we use have different architectures, with the former being a simple rectilinear encoder-decoder and the later being a JumpReLU autoencoder; see the top of page 3 for more information about the SAEs we use.
>
> > For large-scale models with complex circuits, what optimizations or techniques do the authors recommend to maintain interpretability while managing the increased number of features?
>
> The circuit discovery algorithm we use is already relatively scalable for discovering circuits in larger models (assuming that trained SAEs exist for the model). Thus, the remaining challenge is that of interpreting the features. Here, a common technique is to use language models to interpret SAE features; for example, Neuronpedia provides LLM-generated labels for all features in the Gemma-2-2B SAEs we used (see the box in the upper-left at https://www.neuronpedia.org/gemma-2-2b/16-gemmascope-res-16k/0 for an example). While these automatic interpretations are not always reliable, they are a very active area of research and we expect rapid improvement.
>
> Additionally, a subset of the authors have pilot experiments using agents to automatically decide which sparse features in a given circuit are spurious and can be safely ablated. We have found that it is effective to condition on a description of the task when generating the feature labels, and that this improves the agreement between the agent’s and humans’ judgments on which features are spurious for a given task. While we do not yet have fully-formed evidence, our pilot experiments give us hope that we will be able to automate away much of the human labor involved in interpreting the causal mechanisms underlying task-specific behaviors.
>
> References
> ===
> [1] Bills et al. (2023). “Language models can explain neurons in language models.” OpenAI. https://openaipublic.blob.core.windows.net/neuron-explainer/paper/index.html
>
> [2] Lin & Bloom (2023). “Analyzing neural networks with dictionary learning.” Neuronpedia. https://www.neuronpedia.org
>
> [3] Gao et al. (2024). “Scaling and evaluating sparse autoencoders.” https://arxiv.org/abs/2406.04093
>
> [4] Rajamanoharan et al. (2024). “Improving Dictionary Learning with Gated Sparse Autoencoders.” https://arxiv.org/abs/2404.16014
>
> [5] Arora et al. (2024). “CausalGym: Benchmarking causal interpretability methods on linguistic tasks.” ACL. https://aclanthology.org/2024.acl-long.785/

---

> > ### Comment · Reviewer_gvvK · 2024-11-22
> > **Response to authors & Thank you**
> >
> > Thanks for the detailed explanation and clarification. Appreciate it. I'd like to increase my score for this work.

---

### Official Review · Reviewer_SioR · 2024-11-10

**Soundness:** 3
**Presentation:** 3
**Contribution:** 3
**Rating:** 8
**Confidence:** 3

**Summary:**

This paper introduces sparse feature circuits as a causal network built from human-interpretable features, rather than neurons. The author discuss how to quantify the Indirect Effects (IE) of steering or patching individual neurons on model behaviors and propose approximations to extend this approach to models with many components.

Through experiments on the Gemme2-2B and Pythia-128M models, they demonstrate that a relatively low number of sparse features, compared to neurons, can interpret model behaviors with high fidelity. Building on these findings, the authors present the SHIFT framework, which enables the surgical removal of unintended features to prevent them from affecting model classification performance. While the framework involves some human judgment, it highlights that these Sparse Auto-Encoder (SAE) features are effective representations of concepts that can be manipulated to control model behaviors. Finally, the authors propose an unsupervised circuit discovery pipeline, moving from next-token clustering to zero-ablating features, to identify the most important feature circuits.

**Strengths:**

* This paper is well-motivated and articulated;
* The studied subject is of good importance to the NN interpretability domain;
* The experiment setup is clear and the results are promising;
* The idea has sound novelty.

**Weaknesses:**

* SAE features are vast, and identifying specific circuits among those formulated by these features can be challenging.
* SAE errors cannot be fully interpreted, while they play an important role in making sparse feature circuits. This is ongoing work to train more proficient SAEs, but it should be discussed more in the paper regarding how to perform surgical edits on such errors.
* I suspect that the circuits may only correlate with certain features. Simply removing gender-related features and observing a drop in classification performance is not universally convincing as a causal intervention approach; additional experiments on other features are needed to support this claim;
* Trivial concerns: 1) several typos: 283: fair; 201: produce; 492: succession; 2) citation formatting errorsat line 95-96; 3) line 454-457 needs more clarity.

**Questions:**

* Can you add explanations regarding the efficiency of the sparse feature circuit finding with the vast amount of possible feature circuits with respect to neurons'?
* Can you help better understand how to interpret SAE errors at part of the sparse feature circuits?
* Can you share your thoughts on extended experiments on more feature ablating studies and how would the results be?

---

> ### Author Response · Authors · 2024-11-21
> **Response Pt. 1**
>
> Thank you for the feedback! We appreciate that you found the paper to address an important topic and the results to be promising. We address your feedback and questions below.
>
> Weaknesses
> ===
> > …[I]dentifying specific circuits among those formulated by these features can be challenging.
>
> They can indeed. Hence, we propose a causally efficacious technique that scales effectively with the number of components (and combinations thereof). Is the concern regarding the time complexity of the technique, or perhaps the difficulty of locating the exact circuit? We would love to address your concern, so if you have a chance to clarify, we would be happy to discuss further.
>
> > SAE errors cannot be fully interpreted… it should be discussed more in the paper regarding how to perform surgical edits on such errors.
>
> We agree that this is a fundamental limitation of our method. Better SAEs will lead to more interpretable circuits, whereas worse SAEs will put much more of the indirect effects in the error terms.
>
> > …Simply removing gender-related features and observing a drop in classification performance is not universally convincing as a causal intervention approach.
>
> Thank you for the comment. Following prior approaches to studying causality, we establish a causal connection between specific SAE features and the probe's performance by intervening on the SAE features and observing a change in the way the probe classifies gender (which was the predicted effect, since the SAE features ablated were selected for appearing related to gender). If you have remaining concerns about establishing that the circuits identified are causal, would you please elaborate on your concerns? We'd love to address them.
>
> For example, one possible concern might be that ablating *any* SAE features generically interferes with gender classification (but not, for some reason, profession classification). We have added an experiment that addresses this: we now have a baseline for ablating random SAE features instead of SAE features which were selected as being related to gender. We show that this has next to no impact on probe performance (see the revised table 2), indicating that it was the specific SAE features we selected which were having the effect. Please let us know if you would like to see other experiments.
>
> > Other concerns
>
> Thank you for pointing out these formatting errors! These have been fixed in the revised PDF.
>
> Questions
> ===
> > Can you add explanations regarding the efficiency of sparse feature circuit finding with the vast amount of possible feature circuits with respect to neurons?
>
> Please see L139-157. Using the most efficient linear approximation for estimating indirect effects, we can identify causally relevant features using just two forward passes and one backwards pass per input in our dataset. When using the more accurate but less efficient integrated-gradients-based approximation, this grows by a factor of our model’s serial depth (i.e., roughly the number of layers). So overall, identifying the nodes in our circuits is as expensive as inferencing the model over our data a constant number of times; in particular, it does not depend on the number of features. We work with datasets of 100 examples, and this takes a few seconds for both Pythia-70M and Gemma-2-2B.
>
> Computing the edges in our circuits is more expensive: it requires an additional forwards and backwards pass per feature in our circuit (the number of which depends on the chosen threshold for including nodes) and per datapoint. For the circuits we present in this paper, this takes around 10 minutes for Pythia-70M and around one hour for Gemma-2-2B.
>
>  We have added some information discussing the time efficiency of our method.
>
> >  Can you share your thoughts on extended experiments on more feature ablating studies and how would the results be?
>
> As discussed above, we have added experiments for random feature ablations. Are there other experiments that you would be interested in seeing?

---

> ### Author Response · Authors · 2024-11-21
> **Response Pt. 2**
>
> > Can you help better understand how to interpret SAE errors as part of the sparse feature circuits?
>
> Certainly! In this work, we use SAEs to decompose latent activations x as sums \sum_{i=1}^{d_{SAE}} f_i(x) \mathbf{v_i} + \epsilon(x) where the f_i(x) are feature activations and \epsilon(x) is a (high-dimensional) error term. This decomposition allows us to view an LLM as consisting of a computational graph with nodes corresponding to the f_i’s and \epsilon’s. Concretely, you could imagine representing the LLM as a computational graph with a node for each residual stream, attention output, and MLP output; and then breaking each of these nodes apart into d_{SAE} + 1 nodes each. Note that even though these nodes come in two types (nodes representing one-dimensional features and nodes representing high-dimensional errors), there is no difference between these types of nodes from the perspective of the causal structure of the graph. In particular, we can define the causal effect of a node on the model’s output in exactly the same way for these two types of nodes: as the change in the model’s output after intervening on the node by patching it to a counterfactual value. So from a formal perspective, you can view the error terms as components in a circuit in exactly the same way as for sparse features.
>
> However, there is one key difference between sparse features and error terms: it is possible to annotate the nodes corresponding to sparse features with human-understandable interpretations. But this is not possible for the error terms.
>
> We’ll note the causal importance of the error terms, as measured by the change in the model’s training-set perplexity after ablating these errors, was already a central metric of the quality of an SAE prior to this work (though not usually phrased in the language of “error terms”). Thus, better SAEs produce compositions whose error terms are less important for model outputs (and conversely, the interpretable sparse feature nodes explain more of the model’s outputs). You can see this reflected in our work when comparing feature circuits computed with our Pythia-70M SAEs (which have quite important error nodes) to feature circuits computed using the Gemma Scope SAEs for Gemma-2-2B (where the error nodes are substantially less important). One interesting consequence of our work is that—since feature circuits are computed for specific tasks—the weight of error nodes in feature circuits gives a task-specific indication of SAE quality (in contrast to using loss recovered on the full training set).

---

> > ### Comment · Reviewer_SioR · 2024-11-23
> >
> > Thank you for your responses. I believe many of my concerns have been addressed (though I have not yet had the chance to review every detail thoroughly, but I will). There appears to be a consensus among the reviewers, and I see no reason not to raise the score to a strong accept. Excellent work—thank you!

---

### Public Comment · ~Neel_Nanda1 · 2024-11-15
**Paper Thoughts**

Speaking as an unrelated mechanistic interpretability researcher, I think this is **one of the most important mechanistic interpretability papers of the past year, and worth being highlighted at the conference**. I think this paper had two key contributions:

Demonstrating that sparse autoencoders were useful on the downstream task of SHIFT. Much mechanistic interpretability work purely focuses on showing that it produces understanding, and trying to directly argue that this understanding is true, by eg ablating model components predicted to be important, or finding patterns in where a neuron activates. But if we have achieved true understanding, it should enable greater control, and allow us to do things to models that we couldn't easily do before - and if we can't, perhaps our understanding is illusory. Through demonstrating that SAEs could enable useful disentanglement of two concepts on SHIFT, this work provided a new kind of evidence that SAEs were giving some form of real understanding.

On a more zoomed-out level, I think that looking more into applying mech interp to downstream tasks is a healthy direction for keeping the field grounded, and I've been focusing on this in my own research. This paper provided some useful inspiration for me to make that shift.

The second contribution was being the first serious attempt at circuit finding with SAEs. The goal in circuit finding is finding a sparse set of model components that matter for some task, and a key problem in prior work is that often the key "component" is distributed across many heads or neurons. The promise of SAEs was to find the "right" set of monosemantic components to analyse, but no one had properly tested whether it made circuit analysis substantially easier! I don't consider this paper to have conclusively answered that question, but it provided a lot of useful data, and laid useful conceptual and methodological groundwork, like thinking about error nodes and whether to include/exclude them, whether to use integrated gradients over activation patching (and providing a principled translation of integrated gradients into this context), etc. I also consider it impressive that they do circuit finding for Gemma 2 2B, involving dealing with about 6B parameters of SAE weights, which is a fair bit more technically involved than Pythia-70M. This provides valuable data that the method is scalable, though I do expect more efficient approximations to be required to scale it to frontier models.

As empirical evidence of the impact, two of my current ongoing papers are building on this paper, and I often point people to the paper as a guide for the right methods to use when working with SAE circuit finding.

(Disclosure: I was not at all involved in this paper, but I do know the authors and likely have some positive bias due to that. No one asked me to write this)

(Note also: I don't feel confident in the norms here, so feel free to ignore this comment if these kinds of thoughts are not welcome! I don't see other researchers doing this, but I figure that as ICLR has made a deliberate choice to allow public comments during the rebuttal process, they likely want this kind of public feedback from uninvolved researchers, so long as it's constructive)

---

### Meta-Review · Area_Chair_26oh · 2024-12-23

**Metareview:**

Meta-Review

This submission presents a method for discovering and editing interpretable causal structures—referred to as sparse feature circuits—within large language models (LLMs). Previous work in mechanistic interpretability has often focused on more coarse units (e.g., entire attention heads or neurons) that tend to be highly polysemantic. In contrast, this paper leverages recent advances in sparse autoencoders (SAEs) to decompose models into monosemantic features and constructs causal circuits from these interpretable features. By applying causal interventions on these fine-grained circuits, the authors demonstrate applications such as SHIFT, a method for selectively ablating human-identified, task-irrelevant features to improve model generalization and mitigate biases (e.g., gender cues in a profession classification task). Furthermore, the authors scale their approach to discover thousands of feature circuits across diverse behaviors, highlighting the generality and potential impact of their method.

Key Strengths:
	1.	Conceptual Innovation & Importance:
The paper tackles a crucial challenge in mechanistic interpretability—finding a comprehensible “atomic” unit of analysis. SAEs produce monosemantic features, enabling clearer and more faithful circuit-level explanations. This sets the stage for more granular and robust causal analyses than previous neuron- or head-level methods.
	2.	Downstream Utility (SHIFT):
By demonstrating that interpretability improvements can yield practical interventions, the paper moves beyond purely descriptive analyses. The SHIFT experiment shows that once we identify features responsible for spurious correlations (e.g., gender bias in classification), we can remove them to improve generalization without requiring extra supervision. This strongly supports the claim that better understanding leads to actionable control.
	3.	Scalability and Automation:
The authors show that their approach extends to large models and can discover thousands of circuits without human-specified tasks, relying only on an unsupervised pipeline. This advances the field toward automated mechanistic interpretability that can systematically cover broad model behaviors rather than a small set of human-curated phenomena.
	4.	Strong Empirical and Qualitative Evidence:
Multiple reviewers praised the clarity and breadth of the experiments. The results include direct causal tests, human annotations of interpretability, and comparisons to neuron-based approaches. The paper’s extensive supplementary materials, along with a public demo, enhance reproducibility and transparency.

**Additional Comments On Reviewer Discussion:**

Concerns & Responses:
	•	Human Annotation & Scalability:
Reviewers noted that current methods rely on human judgment to identify spurious or irrelevant features. The authors acknowledge this limitation and point to ongoing work on automated feature labeling using LLMs, as well as an encouraging pilot demonstration of automated labeling approaches. Although not fully resolved, the paper provides a clear path for future improvement.
	•	Complexity of Tasks & Generalization:
While most evaluations focus on syntactic agreement and a bias task, the authors’ unsupervised pipeline highlights that the framework is not limited to simple phenomena. They emphasize ongoing work to apply these techniques to more complex tasks and point to their capacity to handle large model behaviors automatically. The introduction of SHIFT already demonstrates the utility of the method in a challenging semantic classification setting.
	•	Quality of SAEs:
The paper relies on well-trained SAEs, and the authors are forthcoming about the importance of SAE quality. However, they note that current SAE training methods are steadily improving, and future SAE innovations will likely enhance the generality and interpretability of discovered features.

Reviewers’ Assessments:

All reviewers provided positive evaluations, pointing to the paper’s robust contributions, clarity, and methodological innovation. Discussions were constructive, and authors satisfactorily addressed queries and concerns. Multiple reviewers raised their scores after the rebuttal, noting that the responses were thorough and addressed their questions regarding human annotations, SAE dependence, and scalability.

Recommendation:

This paper represents a valuable advance in mechanistic interpretability, offering a principled methodology, rigorous experiments, and an impactful application (SHIFT) that demonstrates real-world utility. The convergence of all reviewers’ comments, the paper’s novelty, thoroughness, and potential to inspire further work in both interpretability and controllability strongly support a strong accept decision.

---

### Decision · Program_Chairs · 2025-01-22

Accept (Oral)